# Make Your LVLM KV Cache More Lightweight

**Xihao Chen**  *chenxihao@u.nus.edu*
*Integrative Sciences and Engineering Programme, National University of Singapore*
*School of Computing, National University of Singapore*

**Yangyang Guo**[*]  *guoyang.eric@gmail.com*
*School of Computing, National University of Singapore*

**Roger Zimmermann**  *dcsrz@nus.edu.sg*
*School of Computing, National University of Singapore*

**Reviewed on OpenReview:** *https://openreview.net/forum?id=n77IeySrQl*

## Abstract

Key-Value (**KV**) cache has become a *de facto* component of modern Large Vision-Language Models (**LVLM**s) for inference. While it enhances decoding efficiency in Large Language Models (**LLMs**), its direct adoption in LVLMs introduces substantial GPU memory overhead due to the large number of vision tokens processed during the prefill stage. To tackle this problem, we propose LightKV, a novel approach that reduces KV cache size by exploiting the redundancy among vision-token embeddings. Guided by text prompts, LightKV employs cross-modality message passing to aggregate informative messages across vision tokens and progressively compress them during prefill. This prompt-aware guidance distinguishes our method from prior vision-only compression strategies. We evaluate LightKV on eight open-source LVLMs across eight public benchmark datasets, *e.g.*, MME and SeedBench. Experimental results demonstrate that with only 55% of the original vision tokens, LightKV (a) halves the vision-token KV cache size, (b) reduces computation by up to 40%, and (c) preserves general-purpose performance while significantly outperforming existing baselines. Our code is publicly available at https://github.com/howtoosee/LightKV.

## 1 Introduction

Benefiting from the rapid advancements in Large Language Models (LLMs) (Vicuna Team, 2023; OpenAI, 2024; Llama Team, 2024), Large Vision-Language Models (LVLMs) (Alayrac et al., 2022; Li et al., 2023b; Dai et al., 2023; Bai et al., 2023; Liu et al., 2023a; 2024b;c; Lu et al., 2024; Chen et al., 2024d;c; Wang et al., 2025; Chen et al., 2025) have recently garnered extensive attention. For example, LLaVA (Liu et al., 2023a) and DeepSeek-VL (Lu et al., 2024) have achieved impressive performance on a multitude of general-purpose multi-modal benchmarks (Fu et al., 2024; Yu et al., 2024; Li et al., 2023c). However, the efficiency of LVLMs remains a significant bottleneck for researchers and practitioners in resource-constrained environments.

Key-Value (**KV**) cache (Pope et al., 2023; Kwon et al., 2023) serves as a fundamental technique in optimizing the inference efficiency of mainstream LLMs and LVLMs. However, although KV caching improves inference speed without compromising model performance, it substantially increases GPU memory consumption. This limitation is especially severe with longer sequences generated (Yang et al., 2024; Liu et al., 2024a; Li et al., 2024d). To alleviate this issue, some training-based methods, such as MQA (Hu et al., 2025) and GQA (Ainslie et al., 2023), introduce the sharing of keys and values across different attention heads. As such, the overall KV cache size is accordingly reduced. These approaches, however, suffer from the requirement of heavy model retraining. In contrast, other methods, such as H2O (Zhang et al., 2023b), MiniCache (Liu et al., 2024a), and ElasticCache (Liu et al., 2024d) focus on pruning tokens within the KV cache *during inference* after the

---

[*]Corresponding author.

prefill stage. These methods offer greater flexibility and can be seamlessly applied to existing decoder-only LVLM models with minimal degradation in performance. *Given this, our work primarily focuses on the reduction of vision tokens during inference time.*

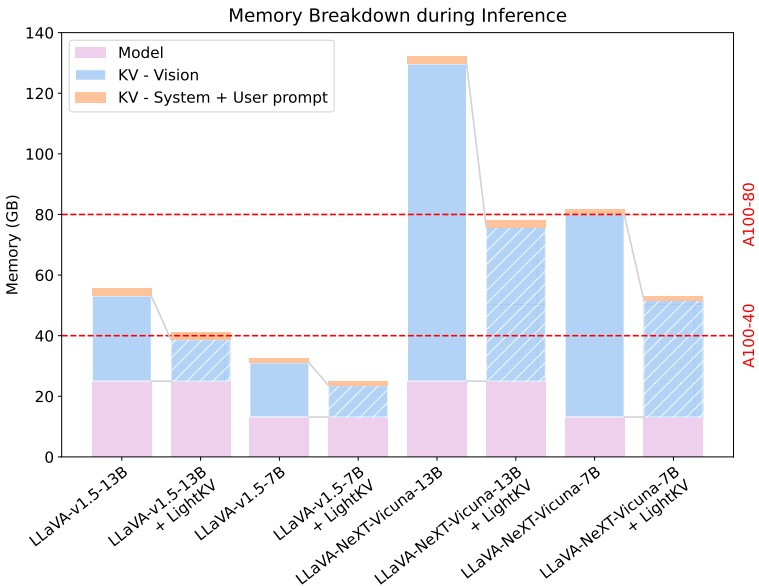

Figure 1: Breakdown of memory consumption in LLaVA models during prefill shows the substantial reduction in KV cache usage with LightKV. As LLaVA-NeXT uses approximately $4\times$ the *vision tokens* as LLaVA-v1.5, there is a sharp increase in memory consumption.

Unlike LLMs, reducing the cost of memory-bound KV cache is challenging in LVLMs due to the following two factors: (a) Tokens in LVLMs are heterogeneous, representing both image patches and text. Determining which tokens should be pruned thus becomes more difficult; (b) The number of tokens computed during the *prefill stage* is significantly larger than that in LLMs. Each image or video frame in LVLMs is embedded into hundreds to thousands of tokens upfront (*e.g.*, 576 in LLaVA-v1.5 (Liu et al., 2023a) and 7,290 in LLaVA-OneVision (Li et al., 2024a)), a considerable amount compared to the context lengths of LLMs (see Fig. 1) (Llama Team, 2024; Jiang et al., 2023; Vicuna Team, 2023). As a result, current LVLMs are limited by significantly heavier GPU memory usage than their LLM counterparts during prefill. A few recent studies have proposed addressing the first challenge on token heterogeneity (Chen et al., 2024a; Li et al., 2024c). However, existing research on solving the second challenge remains sparse.

In this paper, we propose LightKV, a novel method for optimizing KV cache storage in LVLMs during the prefill stage **without retraining**. To this end, we leverage cross-modal prompt guidance to compress vision tokens. Our method follows a three-step design. *First*, we conceptually map each vision token to a graph node, constructing a bipartite graph with edges representing a feature divergence (FD) metric between the connected nodes. Nonetheless, computing FD in a pairwise manner is still expensive, especially with a large number of vision tokens. To alleviate this problem, *second*, we split the vision tokens into sub-windows based on their original spatial locations. This allows us to reduce the complexity of computing FD and aggregating information across tokens, thus improving efficiency. *Third*, our method does not follow existing studies (Chen et al., 2024b) to perform vision token reduction independently, as the text prompts offer more informative signals for vision token importance. Consequently, we leverage on-the-fly cross-modal attention scores between vision tokens and prompt tokens for informed token updates. We find that although this approach has been largely ignored by the existing literature, it delivers superior results to state-of-the-art baselines.

We apply LightKV to eight state-of-the-art LVLM models: LLaVA-v1.5-13B, LLaVA-v1.5-7B (Liu et al., 2023a), LLaVA-NeXT-13B, LLaVA-NeXT-7B (Liu et al., 2024b), InternVL2-8B (Chen et al., 2024c), EVE-7B-v1, EVE-7B-v1-HD (Diao et al., 2025), Qwen2.5-VL (Bai et al., 2025), and conduct extensive experiments across eight benchmarks, *e.g.*, MME (Fu et al., 2024) and SeedBench (Li et al., 2024b). Our results

demonstrate that LightKV can reduce the KV memory of vision tokens by 50% while maintaining, sometimes even surpassing, the vanilla LVLM performance. Furthermore, when constrained with the same token length generation budget, the inference cost in FLOPs is reduced by 40%.

In summary, LightKV reduces the KV cache footprint in LVLMs by compressing vision tokens during the *prefill* stage under the guidance of text prompts. This prompt-aware design distinguishes it from existing SOTA vision-only methods, delivering (1) greater efficiency and (2) superior benchmark performance. Importantly, LightKV is entirely *training-free* and can be seamlessly applied to a wide range of LVLMs, including both vision encoder-based and encoder-free models.

## 2 Related work

**Large vision-language models**  Following the success of large language models (LLMs) in the language domain (Vicuna Team, 2023; OpenAI, 2024; Llama Team, 2024), large vision-language models (LVLMs) have shown substantial progress on various multimodal tasks (Team, 2024b;a; Driess et al., 2023). Current LVLMs primarily fall into the following three directions: (a) Fusion-based methods directly inject vision information into the LLM decoders using cross-attention (Alayrac et al., 2022; Awadalla et al., 2023; Li et al., 2023a; Gong et al., 2023). (b) Query-based LVLMs extract vision information with learnable query tokens, which are then concatenated with text tokens (Li et al., 2023b; Dai et al., 2023; Zhu et al., 2024; Li et al., 2024c; Zhang et al., 2023a). (c) Projection-based methods directly map the encoded tokens from a vision encoder into the text space (Liu et al., 2023a; 2024b;c; Li et al., 2024a; Bai et al., 2023; Huang et al., 2023; Diao et al., 2025). However, despite their simplicity, such a projection substantially increases the memory footprint of the input sequence.

**KV cache optimization**  KV cache has been widely used in LLMs and LVLMs to improve their inference efficiency (Dao et al., 2022; Pope et al., 2023; Kwon et al., 2023; Lee et al., 2024). The core idea is to store the key and value tokens to reduce future redundant computations. However, in situations with long contexts, keeping the KV cache imposes an increased burden on GPU memory. Existing approaches addressing this can be roughly categorized into two groups: (a) KV-sharing-based and (b) token-reduction-based. Specifically, methods from (a) improve the multi-headed attention mechanism to achieve efficiency. For instance, MQA (Hu et al., 2025) and GQA (Ainslie et al., 2023) share keys and values across attention heads (Vaswani et al., 2017), reducing the amount of KV needed to be cached. In contrast, methods from (b) improve KV cache size by pruning or merging tokens based either on minimal importance (Zhang et al., 2023b; Li et al., 2024d; Cai et al., 2024) or attention consistency across layers (Liu et al., 2023b; 2024d; Yang et al., 2024). Beyond LLMs, some initial efforts have been devoted to optimizing the KV cache for LVLMs. In particular, LLaVolta (Chen et al., 2024a), IVTP (Huang et al., 2024) and FastV (Chen et al., 2024b) propose pruning vision tokens in the LLM decoder backbone. The first two require model retraining; FastV, though training-free, prunes vision tokens without cross-modality guidance, yielding inconsistent results across models and benchmarks. In contrast, LightKV leverages guidance from text tokens to deliver more consistent and superior performance across a diverse set of benchmarks.

**Vision token compression**  Tokens in vision transformers (ViTs) (Dosovitskiy et al., 2021) often exhibit high redundancy (Bolya et al., 2023; Pan et al., 2022; Chen et al., 2024b). To address this, some approaches train modules to identify and discard less important tokens (Rao et al., 2021; Bonnaerens & Dambre, 2023; Yin et al., 2022; Fayyaz et al., 2022; Wei et al., 2023; Chen et al., 2023; Zhang et al., 2024; Mao et al., 2025). Some other typical methods first group tokens based on similarity or distance (Bolya et al., 2023; Tran et al., 2024; Kim et al., 2024; Alvar et al., 2025) or image segmentation (Xu et al., 2022; Lu et al., 2023) and then prune or merge the tokens with the maximum similarity. These methods either (a) require the training of additional module(s), or (b) do not support the vision-language joint reasoning as in LVLMs.

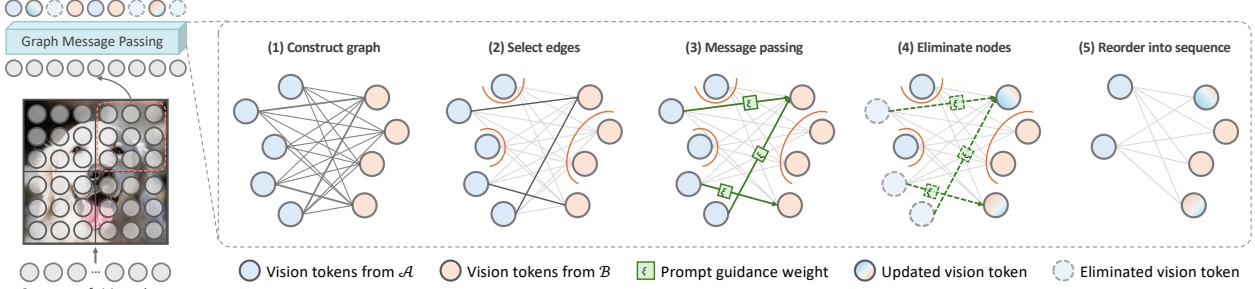

Figure 2: Method overview of intra-window token compression. **Step 1**: Construct a bipartite graph by partitioning the vision tokens into non-overlapping sets $\mathcal{A}$ (blue) and $\mathcal{B}$ (orange), weight each edge by an FD metric, defined in Eq. 5. **Step 2**: Select edges with the smallest $\lfloor \rho v/2 \rfloor$ FD values and delete the rest. The unconnected nodes are left unchanged. **Step 3**: Pass messages from nodes in $\mathcal{A}$ to connected nodes in $\mathcal{B}$, weighted by their corresponding attention scores $\xi$, as computed in Eq. 7. Then, aggregate messages and update nodes in $\mathcal{B}$. **Step 4**: Eliminate the now-redundant nodes from $\mathcal{A}$. **Step 5**: Reorder the remaining nodes into a sequence of vision tokens, serving as input to the next decoder layer.

## 3 Method

### 3.1 Preliminaries

Recent LLMs often operate in an autoregressive fashion: given a sequence of $p$ text prompt tokens $[x_1, \ldots, x_p]$ (including both system prompt and user prompt), and $t - p$ previously generated tokens $[x_{p+1}, \ldots, x_t]$, an LLM with parameters $\Theta$ predicts the next token $x_{t+1}$ with:

$$\mathbb{P}_\Theta\big(x_{t+1} \mid \underbrace{x_1, \ldots, x_p}_{\text{Prompt tokens}}, \underbrace{x_{p+1}, \ldots, x_t}_{\text{Generated tokens}}\big). \tag{1}$$

The above process is often implemented in two stages: prefill and generation (Golden et al., 2024). During *prefill*, the model tokenizes all $p$ prompt tokens and computes the queries $Q_p = [\mathbf{q}_1, \mathbf{q}_2, \ldots, \mathbf{q}_p]$, similarly for keys $K_p$ and values $V_p$ (Vaswani et al., 2017). In contrast, during *generation*, when a new token arrives, the model first obtains the query $\mathbf{q}_{t+1}$, key $\mathbf{k}_{t+1}$, and value $\mathbf{v}_{t+1}$ vectors. It then computes the attention matrix by applying $\mathbf{q}_{t+1}$ to the full set of keys $K_{t+1}$:

$$\mathbf{A} = \text{softmax}\left(\mathbf{q}_{t+1} \, K_{t+1}^\top / \sqrt{d_k}\right), \tag{2}$$

where $d_k$ represents the embedding dimension. In practice, the attention output would be a concatenation of matrices $\mathbf{A} = [\mathbf{A}_1, \ldots, \mathbf{A}_H]$ from $H$ independent attention heads.

**KV cache** From the above, we observe that the autoregressive nature of LLMs allows for the previously computed *keys* $K_t$ and *values* $V_t$ to be reused in future time steps during generation. This operation reduces the computational overhead by preventing the recomputation of key and value tokens (Xu et al., 2025). However, an increased consumption of GPU memory is usually induced by the growing size of the KV cache. This is often manifested as: (a) generating lengthy sequences and (b) caching many contexts during prefill. In this work, we primarily focus on improving the second.

**LVLMs** LVLMs build on LLMs by extending their architecture to process visual information. A common paradigm in LVLMs is to first map the split image patches into tokens using ViT-based encoders (Dosovitskiy et al., 2021; Radford et al., 2021; Bao et al., 2022), and then concatenate these tokens with the prompt tokens to form the input sequence. In general, LVLMs generate tokens by conditioning on both text prompt tokens and vision tokens:

$$\mathbb{P}_\Theta\big(x_{t+1} \mid \underbrace{x_1, \ldots, x_p}_{\text{Prompt tokens}}, \underbrace{x_{p+1}, \ldots, x_{p+v}}_{\text{Vision tokens}}, \underbrace{x_{p+v+1}, \ldots, x_t}_{\text{Generated tokens}}\big). \tag{3}$$

We denote $X_v$ as the sequence of $v$ vision tokens in Eq. 3. Similar to LLMs, KV cache is a key component in speeding up inference in LVLMs. In this paper, we focus primarily on compressing vision tokens for two reasons: (a) as shown in Fig. 1, vision tokens greatly outnumber text prompt tokens; (b) preliminary studies showed that reducing text tokens causes severe performance degradation.

## 3.2 LightKV

As illustrated in Fig. 2, the pipeline of LightKV functions as follows: At each specified decoder layer during the prefill stage, given a sequence of vision tokens, we first reconstruct their grid structure as in the original image. These tokens are then partitioned into $w \times w$ small, non-overlapping windows, each containing an equal number of tokens. Within each window, we perform message passing to compress vision tokens, simultaneously reducing KV size and the length of the vision input to the next decoder layer (see Sec. 3.2.1). This is repeated in later layers with larger effective windows to achieve inter-window compression (see Sec. 3.2.2).

### 3.2.1 Intra-window token compression

To address redundancy in vision tokens, we utilize message passing to aggregate information among tokens with low feature divergence (FD) (see Eq. 5), and subsequently eliminate redundant nodes within each window $\omega$. The message passing and update procedure is applied independently to each window. For notational clarity, we omit the subscript $\omega$ and use $v$ to denote the number of tokens in a window in Sec. 3.2.1.

**Graph construction**  We map the vision tokens within each window to a bipartite graph. For notational simplicity, we slightly abuse notation and use $\mathbf{x}$ to denote the embedding of a vision node. **Step 1**: In each window, we first map each token $\mathbf{x}$ to a graph node, with $\mathcal{X} = \{\mathbf{x}|\mathbf{x} \in X_v\}$. Next, we partition the set of nodes into two near-equal subsets, $\mathcal{X}_\mathcal{A}$ and $\mathcal{X}_\mathcal{B}$ (shown in blue and orange, respectively, in Fig. 2), by assigning tokens in an alternating manner: odd-indexed tokens to $\mathcal{X}_\mathcal{A}$ and even-indexed tokens to $\mathcal{X}_\mathcal{B}$. We then construct a bipartite graph between the two subsets with edges $\mathcal{E}$:

$$\mathcal{E} = \mathcal{X}_\mathcal{A} \times \mathcal{X}_\mathcal{B} = \{(\mathbf{x}_\alpha, \mathbf{x}_\beta) \mid \forall \mathbf{x}_\alpha \in \mathcal{X}_\mathcal{A}, \ \forall \mathbf{x}_\beta \in \mathcal{X}_\mathcal{B}\}, \tag{4}$$

where $\times$ denotes set cross product. We modify the FD in (Tran et al., 2024; Wang et al., 2024) to weight each edge in the graph:

$$\mathrm{FD}(\alpha, \beta) = 1 - \frac{\langle \mathbf{x}_\alpha, \mathbf{x}_\beta \rangle}{||\mathbf{x}_\alpha|| \ ||\mathbf{x}_\beta||}, \tag{5}$$

where $\langle \cdot, \cdot \rangle$ denotes the inner product and $|| \cdot ||$ is the $L^2$-norm. **Step 2**: We compute the feature divergence $\mathrm{FD}(\alpha, \beta)$ for all bipartite pairings between $\mathcal{X}_\mathcal{A}$ and $\mathcal{X}_\mathcal{B}$. These pairs are subsequently ranked in ascending order, and we construct the candidate set $\mathcal{T}_\rho$ by selecting the $\lfloor \rho v/2 \rfloor$ pairs with the lowest FD values, where $\rho$ denotes the ratio of tokens removed. Note that one-to-one matching is not enforced in $\mathcal{T}_\rho$: multiple nodes in $\mathcal{X}_\mathcal{A}$ may connect to the same node in $\mathcal{X}_\mathcal{B}$. We then define the adjacency matrix $M \in \{0,1\}^{|\mathcal{X}_\mathcal{A}| \times |\mathcal{X}_\mathcal{B}|}$ as

$$M_{\alpha,\beta} = \begin{cases} 1, & \text{if } (\alpha, \beta) \in \mathcal{T}_\rho, \\ 0, & \text{otherwise.} \end{cases} \tag{6}$$

Edges not in $\mathcal{T}_\rho$ are temporarily removed and unconnected nodes $\mathcal{X}_\mathcal{R} = \{\mathbf{x}_r | \nexists \beta \text{ s.t.} (r, \beta) \in \mathcal{T}_\rho\}$ are unchanged.

**Token message passing**  In LVLMs, the heterogeneity of tokens introduces a challenge in evaluating the importance of each vision token, and prior works often disregard this by compressing tokens uniformly without accounting for their relative significance. Instead, LightKV reuses the attention weights from the LLM decoder to estimate token importance, which are *readily available during prefill without additional computation*, as shown in Eq. 2. This serves as a signal to preserve the visual features most relevant to the prompt, as measured by how strongly each vision token attends to the prompt tokens, and is used as guidance in the message-aggregation process. **Step 3**: Given the $H$-headed attention matrix $A \in \mathbb{R}^{H \times (p+v) \times (p+v)}$, for a vision token with index $i$, we accumulate the attention of each vision token towards the prompt tokens:

$$\xi_i = \sum_{h=1}^{H} \sum_{j \in \mathcal{J}} \mathbf{A}[h, i, j], \tag{7}$$

where $\mathcal{J}$ is the set of indices for the $p$ prompt tokens. Here, $\mathbf{A}[h, i, j]$ denotes the attention weight where the query corresponds to vision token $i$ and the key corresponds to prompt token $j$. Thus, $\xi_i$ captures how strongly each vision token aligns with the prompt semantics. Next, we gather the attention for each window $\omega$ into vectors $\boldsymbol{\xi}_{\mathcal{A}} \in \mathbb{R}^{|\mathcal{X}_{\mathcal{A}}|}$ and $\boldsymbol{\xi}_{\mathcal{B}} \in \mathbb{R}^{|\mathcal{X}_{\mathcal{B}}|}$ with the same partitions as $\mathcal{X}_{\mathcal{A}}$ and $\mathcal{X}_{\mathcal{B}}$. We update $X_{\mathcal{B}}$ by accumulating messages from its adjacent tokens:

$$
X_{\mathcal{B}} = \underbrace{\left( \boldsymbol{\xi}_{\mathcal{B}} + M^{\top} \boldsymbol{\xi}_{\mathcal{A}} \right)^{-1}}_{\text{(3) Normalize by sum of attentions}} \times \Big( \underbrace{X_{\mathcal{B}} \odot \boldsymbol{\xi}_{\mathcal{B}}}_{\text{(1) Prompt-guidance for } \mathcal{B}} + M^{\top} \underbrace{\underbrace{( X_{\mathcal{A}} \odot \boldsymbol{\xi}_{\mathcal{A}} )}_{\text{(1) Prompt-guidance for } \mathcal{A}}}_{\text{(2) Message passing over edges } M} \Big),
$$

where $(\cdot)^{-1}$ denotes element-wise inverse and $\odot$ is the Hadamard product. This can be broken down into three parts: **(1)** Messages from each token $\mathbf{x}_i$ are weighted by its attention $\xi_i$. **(2)** Next, messages from the tokens in $\mathcal{X}_{\mathcal{A}}$ are passed to those in $\mathcal{X}_{\mathcal{B}}$ through the edges defined in $M$, updating tokens in $\mathcal{X}_{\mathcal{B}}$. The choice of direction is arbitrary, and the reverse direction can be defined analogously. **(3)** Finally, tokens in $\mathcal{X}_{\mathcal{B}}$ are normalized to remain scale-invariant.

Importantly, our aggregation operation utilizes the attention $\xi$ as guidance, ensuring the preservation of visual information that is most relevant to the prompt and the generation of the final response. **Step 4**: After the update, the now-redundant nodes in $\mathcal{X}_{\mathcal{A}} \setminus \mathcal{X}_{\mathcal{R}}$ are deleted. **Step 5**: Finally, the unchanged tokens $\mathcal{X}_{\mathcal{R}}$ and the updated $\mathcal{X}_{\mathcal{B}}$ are concatenated to form the final sequence of tokens for window $\omega$.

**Complexity**  In contrast to computing fully pairwise FD among $v$ vision tokens in each window (which requires $\frac{1}{2}v(v-1)$ computations), the bipartite strategy reduces this by half to $\sim \frac{1}{4}v^2$. We further validate this lower cost empirically in Table 13.

**Difference from ToMe**  LightKV adopts a bipartite matching approach, similar to ToMe (Bolya et al., 2023), to reduce the cost of pairwise calculations. However, ToMe and subsequent methods assume all tokens are equally important, merging them without differentiation. In contrast, LightKV uses cross-modality attention to guide message passing and aggregation, preserving the most relevant information during compression, yielding superior results (see Sec. 4).

### 3.2.2 Inter-window token compression

In this section, the subscript $\omega$ is used to denote variables specific to an individual spatial window.

**Window partitioning**  As discussed above, we split the entire set of vision tokens into window partitions in a non-overlapping manner. Specifically, each window $\omega$ contains $v_\omega = v/(w \times w)$ vision tokens. This reduces the total number of operations involved in computing FD measures from the original $\frac{1}{2}v(v-1)$ to $\frac{1}{2}\frac{v}{w^2}(\frac{v}{w^2}-1) \times w^2 \to \frac{1}{2}v(\frac{v}{w^2}-1)$. Moreover, since spatially adjacent patches typically share semantic similarities, our window-based method confines message aggregation to within a small locality, preserving the positional information of tokens in the original image (Song et al., 2024; Norouzi et al., 2024). A global message passing strategy might inadvertently aggregate information from tokens representing unrelated entities, compromising locality and semantic coherence (Xu et al., 2022; Pan et al., 2022).

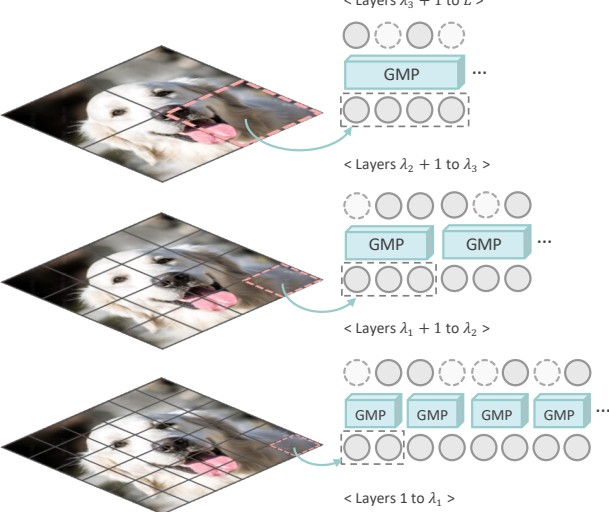

Figure 3: After each compression step, $w$ is reduced to allow message passing across greater spatial distances.

**Hierarchical structure**  We adopt a *hierarchical* compression strategy to improve efficiency, inspired by Swin-Transformer (Liu et al., 2021). Prior studies have shown that LLMs and LVLMs exhibit a layer-wise

semantic hierarchy, where earlier layers tend to capture more local semantics while later layers progressively encode more global relations (Du et al., 2025; Li et al., 2026). Motivated by this observation, we design an iterative vision-token compression strategy that combines intra-window compression in earlier stages with progressively broader inter-window information aggregation in later stages. Given an LVLM with $L$ layers, we perform $s$ compression iterations (where $s < L$), governed by three scheduling hyperparameters:

$$\Lambda = [\lambda_1, \dots, \lambda_s], \quad \mathcal{W} = [w_1, \dots, w_s], \quad \mathcal{P} = [\rho_1, \dots, \rho_s].$$

The hyperparameters $\Lambda$, $\mathcal{W}$, and $\mathcal{P}$ define the layer, window partition, and compression schedules, respectively. For the $i$-th iteration, $\lambda_i$ denotes the target decoder layer, $w_i^2$ represents the number of window partitions, and $\rho_i$ specifies the per-step ratio in token reduction. Specifically, the vision tokens exiting the decoder layer $\lambda_i$ are partitioned into $w_i^2$ partitions. Within each window, vision tokens are compressed such that only a fraction $(1 - \rho_i)$ remains for subsequent layers. By enforcing $w_i > w_{i+1}$, we progressively expand the spatial scope of message passing across iterations, thus achieving hierarchical compression shown in Fig. 3.

## 3.3 Complexity analysis

Without any compression, the prefill stage processes in total $v \times L$ vision tokens.[1] With $s$ compression steps, the number of vision tokens processed during prefill now reduces to:

$$v \times \left( \underbrace{\lambda_1}_{(1)} + \underbrace{\sum_{i=2}^{s} \left( (\lambda_i - \lambda_{i-1}) \prod_{j=1}^{i-1} (1 - \rho_j) \right)}_{(2)} + \underbrace{(L - \lambda_s) \prod_{j=1}^{s} (1 - \rho_j)}_{(3)} \right). \tag{8}$$

If we consider the number of vision tokens in each layer independently, then the total number of vision tokens processed in $L$ decoder layers in a vanilla LVLM is simply $v \times L$. However, the number of vision tokens reduces at every compression layer $\lambda_i$ (note that message passing and accumulation occur after each decoder layer $\lambda_i$). $v \times \prod_{j=1}^{i-1} (1 - \rho_j)$ denotes the number of remaining vision tokens after $i - 1$ accumulation steps. Then, between each pair of accumulation layers $\lambda_{i-1}$ and $\lambda_i$, the number of vision tokens processed is $\left( v \times \prod_{j=1}^{i-1} (1 - \rho_j) \right) \times (\lambda_i - \lambda_{i-1})$. Therefore, Eq. 8 can be broken down into: (1) number of vision tokens processed before the first accumulation step, (2) number of vision tokens processed between the first and the last accumulation step, and (3) number of vision tokens processed after the last accumulation step. For example, in an LVLM with $L = 40$ decoder layers, choosing $\Lambda = [10, 20, 30]$ and $\mathcal{P} = [0.5, 0.5, 0.5]$ reduces the vision token count to 46.9% of the baseline.

# 4 Experiments

## 4.1 Experimental settings

**LVLM base models** We evaluated the efficiency and performance of LightKV by applying it to eight open-source LVLMs: LLaVA-v1.5-13B, LLaVA-v1.5-7B, LLaVA-NeXT-13B, LLaVA-NeXT-7B, InternVL2-8B, EVE-7B-v1, EVE-7B-v1-HD, and Qwen2.5-VL-7B-Instruct. LLaVA-v1.5 encodes 576 vision tokens per image, while LLaVA-NeXT uses 2,144. In contrast, InternVL2 and Qwen2.5-VL adopt dynamic vision encoding, with token counts determined by image resolution. It is worth noting that, unlike other models, which employ a dedicated image encoder, EVE is vision encoder-free. These base models are labeled as *Vanilla*.

**Datasets** We utilized eight publicly available large-scale benchmark datasets for evaluation: Coco Caption (Lin et al., 2014), GQA (Hudson & Manning, 2019), MME (Fu et al., 2024), NoCaps (labeled "NC") (Agrawal et al., 2019), Pope (Li et al., 2023c), SeedBench ("Seed") (Li et al., 2024b), ScienceQA ("SQA") (Lu et al., 2022), and VizWiz ("VW") (Gurari et al., 2018). These benchmarks cover a wide range of tasks, from general, everyday image understanding to fine-grained image reasoning. MME, Pope, SeedBench, and ScienceQA are limited to single-choice answers, while Coco Caption, GQA, NoCaps, and VizWiz involve open-ended responses comprising long sentences.

---

[1] We omit the double estimation of key and value cache for simplicity.

Table 1: Results of LightKV on LLaVA models at 55% vision token retention in the KV cache. **Avg %** denotes the average of all performance metrics normalized against the Vanilla model. Methods are grouped by category and sorted by average score. "NC" and "VW" denote NoCaps and VizWiz, respectively.

| | Method | FLOPs | Mem | TTFT | Coco | MME | | NC | Pope | | Seed | VW | Avg % |
|---|---|---|---|---|---|---|---|---|---|---|---|---|---|
| | | Tera ↓ | GB ↓ | sec ↓ | | C | P | | Acc | F1 | | | |
| **LLaVA-v1.5-13B** | Vanilla | 19.4 | 0.55 | 0.130 | 1.16 | 295.4 | 1532.0 | 1.09 | 0.87 | 0.86 | 0.69 | 0.57 | 100.00 |
| | **Post prefill** | | | | | | | | | | | | |
| | Elastic | 19.3 | 0.31 | 0.598 | 0.96 | 295.4 | 1534.5 | 0.87 | 0.43 | 0.96 | OOM | 0.14 | 68.54 |
| | Rand | 19.0 | 0.31 | 0.134 | 0.48 | 295.4 | 1532.9 | 0.46 | 0.46 | 0.89 | 0.70 | 0.13 | 70.53 |
| | ImgRand | 19.0 | 0.31 | 0.134 | 0.95 | 295.4 | 1532.9 | 0.86 | 0.69 | 0.91 | 0.70 | 0.19 | 85.09 |
| | ToMe (C) | 19.0 | 0.33 | 0.141 | 1.00 | 295.4 | 1532.9 | 0.92 | 0.79 | 0.88 | 0.70 | 0.18 | 87.10 |
| | **During prefill** | | | | | | | | | | | | |
| | ToFu | 12.6 | 0.37 | 0.094 | 1.14 | 292.1 | 1535.7 | 1.08 | 0.86 | 0.86 | 0.38 | 0.55 | 93.36 |
| | PiToMe | 12.6 | 0.37 | 0.093 | 1.14 | 297.5 | 1529.0 | 1.07 | 0.87 | 0.85 | 0.38 | 0.55 | 93.42 |
| | ToMe (P) | 12.6 | 0.37 | 0.094 | 1.16 | 297.5 | 1529.9 | 1.07 | 0.87 | 0.86 | 0.39 | 0.55 | 93.96 |
| | LightKV | 12.6 | 0.37 | 0.098 | 1.15 | 302.1 | 1543.8 | 1.08 | 0.87 | 0.86 | 0.69 | 0.56 | 99.94 |
| | FastV | 12.4 | 0.36 | 0.085 | 1.16 | 308.9 | 1546.6 | 1.09 | 0.86 | 0.85 | 0.68 | 0.57 | **100.22** |
| **LLaVA-v1.5-7B** | Vanilla | 10.2 | 0.35 | 0.078 | 1.10 | 355.7 | 1509.6 | 1.05 | 0.87 | 0.86 | 0.66 | 0.54 | 100.00 |
| | **Post prefill** | | | | | | | | | | | | |
| | Elastic | 10.2 | 0.20 | 0.449 | 0.41 | 350.4 | 1508.9 | 0.30 | 0.30 | 0.93 | OOM | 0.09 | 52.95 |
| | Rand | 9.9 | 0.21 | 0.081 | 0.13 | 350.4 | 1508.9 | 0.10 | 0.74 | 0.87 | 0.66 | 0.11 | 65.80 |
| | ToMe (C) | 10.0 | 0.20 | 0.086 | 0.13 | 350.4 | 1508.9 | 0.09 | 0.87 | 0.86 | 0.66 | 0.18 | 69.02 |
| | ImgRand | 9.9 | 0.20 | 0.082 | 0.22 | 350.4 | 1508.9 | 0.16 | 0.86 | 0.86 | 0.66 | 0.16 | 70.27 |
| | **During prefill** | | | | | | | | | | | | |
| | HiRED | - | - | - | 1.03 | 335.0 | 1452.0 | 1.00 | 0.85 | 0.83 | 0.66 | 0.53 | 96.45 |
| | ToMe (P) | 6.6 | 0.23 | 0.058 | 1.09 | 319.6 | 1490.5 | 1.01 | 0.87 | 0.86 | 0.66 | 0.52 | 97.52 |
| | PiToMe | 6.6 | 0.23 | 0.058 | 1.08 | 341.0 | 1498.5 | 1.02 | 0.86 | 0.85 | 0.65 | 0.51 | 97.63 |
| | ToFu | 6.6 | 0.23 | 0.058 | 1.09 | 340.0 | 1482.3 | 1.02 | 0.86 | 0.85 | 0.66 | 0.52 | 97.98 |
| | FastV | 5.3 | 0.22 | 0.052 | 1.10 | 351.1 | 1513.7 | 1.04 | 0.85 | 0.83 | 0.66 | 0.54 | 99.03 |
| | LightKV | 6.6 | 0.23 | 0.065 | 1.11 | 357.5 | 1519.8 | 1.03 | 0.87 | 0.86 | 0.66 | 0.53 | **99.79** |
| **LLaVA-NeXT-13B** | Vanilla | 65.0 | 1.75 | 0.656 | 1.02 | 318.9 | 1575.1 | 0.88 | 0.88 | 0.86 | 0.69 | 0.64 | 100.00 |
| | **Post prefill** | | | | | | | | | | | | |
| | Elastic | - | - | 2.302 | OOM | OOM | OOM | OOM | OOM | OOM | OOM | OOM | 0.00 |
| | Rand | 60.8 | 0.91 | 0.651 | 0.06 | 318.9 | 1575.1 | 0.04 | 0.82 | 0.86 | 0.69 | 0.08 | 64.51 |
| | ToMe (C) | 61.3 | 0.93 | 0.683 | 0.07 | 318.9 | 1575.1 | 0.05 | 0.87 | 0.86 | 0.69 | 0.08 | 65.48 |
| | ImgRand | 60.8 | 0.91 | 0.652 | 0.07 | 318.9 | 1575.1 | 0.05 | 0.87 | 0.86 | 0.69 | 0.08 | 65.50 |
| | **During prefill** | | | | | | | | | | | | |
| | ToMe (P) | 37.3 | 1.05 | 0.394 | 0.97 | 308.5 | 1551.0 | 0.84 | 0.87 | 0.86 | 0.34 | 0.60 | 90.96 |
| | ToFu | 37.3 | 1.05 | 0.394 | 0.97 | 305.0 | 1539.5 | 0.83 | 0.88 | 0.87 | 0.36 | 0.60 | 91.31 |
| | PiToMe | 37.3 | 1.05 | 0.396 | 0.98 | 311.9 | 1558.2 | 0.86 | 0.87 | 0.86 | 0.34 | 0.60 | 91.56 |
| | FastV | 36.1 | 1.04 | 0.321 | 0.91 | 311.1 | 1477.5 | 0.81 | 0.82 | 0.78 | 0.68 | 0.61 | 93.80 |
| | LightKV | 37.3 | 1.05 | 0.383 | 0.96 | 326.1 | 1576.5 | 0.83 | 0.87 | 0.86 | 0.69 | 0.61 | **98.12** |
| **LLaVA-NeXT-7B** | Vanilla | 34.8 | 1.12 | 0.397 | 1.00 | 330.0 | 1528.2 | 0.88 | 0.88 | 0.86 | 0.68 | 0.61 | 100.00 |
| | **Post prefill** | | | | | | | | | | | | |
| | Elastic | 34.7 | 0.58 | 1.693 | 0.02 | 332.1 | 1519.3 | 0.01 | 0.18 | 0.90 | OOM | 0.08 | 42.67 |
| | Rand | 32.2 | 0.58 | 0.397 | 0.02 | 322.5 | 1523.2 | 0.01 | 0.65 | 0.87 | 0.68 | 0.08 | 61.08 |
| | ImgRand | 32.2 | 0.58 | 0.397 | 0.02 | 322.5 | 1523.2 | 0.02 | 0.85 | 0.87 | 0.68 | 0.08 | 64.06 |
| | ToMe (C) | 32.5 | 0.60 | 0.416 | 0.03 | 322.5 | 1523.2 | 0.02 | 0.87 | 0.86 | 0.68 | 0.08 | 64.33 |
| | **During prefill** | | | | | | | | | | | | |
| | FastV | 18.5 | 0.65 | 0.197 | 0.88 | 265.4 | 1341.3 | 0.78 | 0.81 | 0.77 | 0.69 | 0.58 | 90.37 |
| | HiRED | - | - | - | 0.73 | 297.9 | 1398.9 | 0.67 | 0.88 | 0.87 | 0.66 | 0.58 | 90.68 |
| | ToMe (P) | 21.1 | 0.67 | 0.245 | 0.93 | 292.9 | 1419.0 | 0.78 | 0.88 | 0.87 | 0.65 | 0.57 | 94.18 |
| | ToFu | 20.0 | 0.67 | 0.245 | 0.93 | 295.4 | 1427.2 | 0.78 | 0.88 | 0.87 | 0.66 | 0.57 | 94.52 |
| | PiToMe | 20.0 | 0.67 | 0.247 | 0.94 | 292.1 | 1415.5 | 0.79 | 0.88 | 0.87 | 0.65 | 0.58 | 94.58 |
| | LightKV | 22.3 | 0.67 | 0.259 | 0.98 | 338.6 | 1517.3 | 0.83 | 0.88 | 0.86 | 0.69 | 0.58 | **98.85** |

**Compared baselines**  We adapted two existing techniques from other related domains: ToMe (Bolya et al., 2023) (labeled "*ToMe (C)*") and *ElasticCache* (Liu et al., 2024d). For comparison, we implemented two

Table 2: Results of LightKV on InternVL2-8B at two vision token retention rates in KV cache. "**Avg %**" denotes the average of all metrics normalized against the Vanilla model. Methods are sorted by average score. "VW" denotes VizWiz.

| Method | FLOPs | Mem | TTFT | Coco | GQA | MME | | Pope | | SQA | VW | Avg % |
|--------|-------|-----|------|------|-----|-----|---|------|---|-----|-----|-------|
| | *Tera* ↓ | *GB* ↓ | *sec* ↓ | | | C | P | Acc | F1 | | | |
| Vanilla | 35.7 | 0.24 | 0.460 | 0.90 | 0.63 | 587.5 | 1623.8 | 0.88 | 0.87 | 0.97 | 0.61 | 100.00 |
| **During prefill, retain 60% vision tokens** | | | | | | | | | | | | |
| FastV | 24.8 | 0.15 | 0.520 | 0.80 | 0.50 | 569.6 | 1610.9 | 0.47 | 0.87 | 0.49 | 0.53 | 81.90 |
| ToFu | 22.1 | 0.15 | 0.395 | 0.81 | 0.62 | 502.1 | 1575.5 | 0.87 | 0.86 | 0.94 | 0.60 | 95.49 |
| PiToMe | 22.1 | 0.15 | 0.396 | 0.99 | 0.60 | 461.8 | 1545.3 | 0.87 | 0.86 | 0.90 | 0.60 | 95.99 |
| ToMe (P) | 22.1 | 0.15 | 0.397 | 0.87 | 0.62 | 551.4 | 1621.8 | 0.87 | 0.86 | 0.95 | 0.60 | 97.86 |
| LightKV | 23.1 | 0.15 | 0.391 | 0.91 | 0.63 | 590.0 | 1623.8 | 0.88 | 0.87 | 0.97 | 0.61 | **100.19** |
| **During prefill, retain 55% vision tokens** | | | | | | | | | | | | |
| FastV | 22.9 | 0.14 | 0.517 | 0.68 | 0.47 | 582.1 | 1611.1 | 0.56 | 0.85 | 0.46 | 0.48 | 79.49 |
| PiToMe | 22.1 | 0.15 | 0.396 | 1.00 | 0.61 | 442.9 | 1575.5 | 0.87 | 0.86 | 0.90 | 0.57 | 95.54 |
| ToMe (P) | 22.1 | 0.15 | 0.397 | 0.81 | 0.62 | 503.9 | 1570.0 | 0.87 | 0.86 | 0.95 | 0.60 | 95.62 |
| ToFu | 22.1 | 0.15 | 0.395 | 0.75 | 0.62 | 541.8 | 1619.1 | 0.87 | 0.85 | 0.95 | 0.60 | 95.82 |
| LightKV | 23.1 | 0.15 | 0.391 | 0.88 | 0.62 | 590.0 | 1623.8 | 0.88 | 0.87 | 0.97 | 0.61 | **99.58** |

Table 3: Results of LightKV on EVE-7B-v1 models at 55% retention of vision tokens in the KV cache. "NC" and "VW" denote NoCaps and VizWiz, respectively.

| Method | Coco | MME | | NC | Pope | | VW | Avg % |
|--------|------|-----|---|-----|------|---|-----|-------|
| | | C | P | | Acc | F1 | | |
| **EVE-7B-v1** | | | | | | | | |
| Vanilla | 0.96 | 269.2 | 1230.8 | 0.94 | 0.84 | 0.83 | 0.46 | 100.00 |
| FastV | 0.85 | 259.3 | 1144.5 | 0.78 | 0.80 | 0.77 | 0.44 | 92.07 |
| LightKV | 1.00 | 269.3 | 1203.1 | 0.93 | 0.84 | 0.83 | 0.43 | **99.20** |
| **EVE-7B-v1-HD** | | | | | | | | |
| Vanilla | 1.05 | 304.6 | 1314.1 | 1.02 | 0.86 | 0.85 | 0.56 | 100.00 |
| FastV | 0.97 | 290.3 | 1238.6 | 0.93 | 0.83 | 0.82 | 0.55 | 94.90 |
| LightKV | 0.97 | 291.4 | 1308.9 | 0.94 | 0.86 | 0.85 | 0.54 | **96.61** |

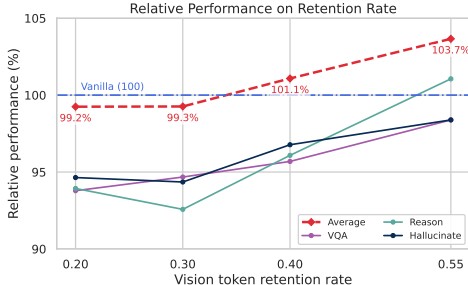

Figure 4: Effect of varying retention rates on Qwen2.5-VL. The "Average" curve summarizes the overall performance trend across Reasoning, VQA, Hallucination and Captioning.[3]

random-eviction baselines: *Rand* and *ImgRand*. *Rand* and *ElasticCache* prune both text and vision tokens, whereas *ImgRand* and ToMe reduce vision tokens only. It is important to note that the previously mentioned methods perform token reduction *after* the prefill stage. Additionally, for token reduction *during* prefill, we implemented ToMe (labeled "*ToMe (P)*") and four recent SOTA strategies: *FastV* (Chen et al., 2024b), *PiToMe* (Tran et al., 2024), *ToFu* (Kim et al., 2024) and *HiRED* (Arif et al., 2025).[2]

**Implementation details**  In our experiments, we retain the default parameters of the LVLM backbones and use greedy decoding for reproducibility. For FastV, we adopt the reported optimal setting of $K = 2$ and vary only $R$ to control the KV cache pruning ratio. For other methods, we adapted them to work with the LVLM backbones as faithfully as possible. To ensure consistency, we fix the schedule of LightKV's compression layers $\Lambda$, compression ratios $\mathcal{P}$, and window sizes $\mathcal{W}$ across all benchmarks for each LVLM model. We utilized *lmms-eval* (Zhang et al., 2025) for all benchmark evaluations. We profiled the time-to-first-token (TTFT) and the generation latency for 100 tokens by averaging over 10 runs on an NVIDIA A100 GPU.

---

[2]HiRED uses the same model but with HuggingFace optimizations; efficiency metrics are omitted for fairness.

[3]The Captioning trend is omitted because its performance remains above 105%, exceeding the current vertical axis range.

Table 4: Comparison of prompt-guided weighting to uniform and random variants at 55% vision token retention.

| Method | Coco | MME | | NC | Pope | | Seed | VW | Avg % |
|--------|------|-----|------|-----|------|------|------|------|-------|
| | | C | P | | Acc | F1 | | | |
| **LLaVA-v1.5-13B** | | | | | | | | | |
| Uniform | 1.14 | 299.6 | 1535.0 | 1.06 | 0.87 | 0.86 | 0.39 | 0.55 | 93.78 |
| Random | 1.14 | 300.0 | 1534.5 | 1.07 | 0.87 | 0.85 | 0.39 | 0.56 | 93.96 |
| Prompt | 1.15 | 302.1 | 1543.8 | 1.08 | 0.87 | 0.86 | 0.69 | 0.56 | **99.94** |
| **LLaVA-NeXT-13B** | | | | | | | | | |
| Uniform | 0.98 | 311.0 | 1547.3 | 0.85 | 0.86 | 0.85 | 0.35 | 0.60 | 91.18 |
| Random | 0.97 | 311.1 | 1542.3 | 0.84 | 0.86 | 0.86 | 0.34 | 0.59 | 90.65 |
| Prompt | 0.96 | 326.1 | 1576.5 | 0.83 | 0.87 | 0.86 | 0.69 | 0.61 | **98.12** |

## 4.2 Main results

We compare the performance of LightKV with other SOTA methods on LLaVA models (Table 1), InternVL (Table 2), EVE (Table 3) and Qwen2.5-VL (Fig. 4 and Table 10 in the appendix). For each LVLM model, we selected the optimal configurations of $\Lambda$ and $\mathcal{W}$ based on performance on Coco and MME, and applied these hyperparameters to the remaining benchmarks. We also profiled efficiency metrics, including FLOPs, KV cache memory (from prompt, vision, and generated tokens), and time to first token (TTFT) when generating 100 tokens (standard deviation reported in the supplementary). Our key findings are summarized as follows:

- Tables 1, 2, 3 and 10 show that LightKV consistently preserves the performance of the base LVLMs across most benchmarks. In some cases, our method surpasses the performance of LVLMs without compression.

- Compared to methods applied *during the prefill stage* (see Table 1), LightKV either outperforms or achieves highly competitive results, ranking first in 3 out of 4 LLaVA models and second in the remaining one. Furthermore, baseline methods often obtain lower FLOPs or memory at the cost of larger performance degradation, whereas LightKV provides a stronger performance-efficiency tradeoff.

- Our method yields the most consistent performance across the models, while others exhibit inconsistent rankings due to substantial degradations. For example, FastV performs well on LLaVA-v1.5 models, but shows substantial drops on LLaVA-NeXT models. We attribute this to its pruning strategy, which removes vision tokens solely based on early-layer visual attention scores. Given that LLaVA-v1.5 encodes only 576 vision tokens while LLaVA-NeXT processes 2,144, early-layer attention in the latter is far sparser and less reliable as an importance signal, causing FastV to prematurely discard tokens that later contribute to cross-modal reasoning, a shortfall mitigated by our hierarchical strategy.

- At even more aggressive compression ratios (*e.g.*, retaining 20% and 30%), LightKV is capable of retaining 99% average performance across multiple benchmarks on Qwen2.5-VL (Fig. 4 and Table 10 in the appendix), further highlighting its robustness.

- LightKV is compatible not only with vision encoder-based LVLMs, but also with encoder-free models such as EVE, which seek to reduce the strong inductive bias in the vision encoders. As shown in Table 3, our approach substantially outperforms FastV at the same compression rate, and is better at preserving the original capabilities of the LVLMs.

- Post-prefill approaches substantially degrade performance on open-ended tasks, *e.g.*, Coco and NoCaps. Additionally, they yield minimal improvements in efficiency, since the prefill stage remains the dominant memory and latency bottleneck. In contrast, LightKV operates during prefill within the decoder layers, resulting in significantly lower compute cost and memory footprint while achieving stronger performance.

Table 5: TTFT (ms) and 100-token generation latency (s) $\pm$ Std. Dev. on LLaVA 13B models.

| Method | TTFT (ms) | Gen latency (s) | TTFT (ms) | Gen latency (s) |
|---|---|---|---|---|
| | **LLaVA-v1.5-13B** | | **LLaVA-NeXT-13B** | |
| Vanilla | $130 \pm 0.393$ | $2.89 \pm 0.004$ | $656 \pm 1.479$ | $3.79 \pm 0.013$ |
| LightKV | $98 \pm 1.045$ | $2.80 \pm 0.004$ | $383 \pm 0.878$ | $3.29 \pm 0.002$ |

Table 6: Effect of varying window sizes $w$ at different compression layers on the performance of InternVL-8B across benchmarks. "VW" denotes VizWiz.

| Method | $\mathcal{W}$ | Coco | GQA | MME | | Pope | | SQA | VW |
|---|---|---|---|---|---|---|---|---|---|
| | | | | C | P | Acc | F1 | | |
| Vanilla | - | 0.90 | 0.63 | 587.5 | 1623.8 | 0.88 | 0.87 | 0.97 | 0.61 |
| LightKV | | | | | $\lambda = 3$ | | | | |
| | 1 | 0.80 | 0.62 | 547.5 | 1602.5 | 0.87 | 0.86 | 0.95 | 0.60 |
| | 2 | 0.83 | 0.59 | 555.0 | 1621.1 | 0.87 | 0.86 | 0.96 | 0.60 |
| | 4 | 0.90 | 0.60 | 546.8 | 1594.8 | 0.87 | 0.85 | 0.95 | 0.60 |
| | | | | | $\lambda = 14$ | | | | |
| | 1 | 0.89 | 0.62 | 577.1 | 1615.8 | 0.87 | 0.86 | 0.97 | 0.61 |
| | 2 | 0.90 | 0.62 | 577.1 | 1620.3 | 0.87 | 0.86 | 0.97 | 0.61 |
| | 4 | 0.92 | 0.62 | 577.9 | 1617.5 | 0.88 | 0.86 | 0.97 | 0.61 |

## 4.3 Additional experiments

**Effect of prompt guidance**   To isolate the specific contribution of prompt-aware guidance within LightKV, we conduct an ablation study comparing our approach against variants utilizing uniform and random attention weights. As reported in Table 4, substituting our prompt-guided mechanism with these simpler weighting schemes results in consistent performance degradation across benchmarks. These findings validate the performance gains stemming from the cross-modal signals during compression.

**Latency profiling**   Table 5 illustrates the reduction in TTFT and generation latency over 100 tokens achieved by LightKV. As our approach requires explicit attention matrices, it is incompatible with I/O-optimized mechanisms like FlashAttention (Dao et al., 2022). To overcome this, we selectively switch to eager computation in the small subset ($s \ll L$) of layers where compression is applied, while retaining the optimized attention implementation for the majority. The marginal overhead is offset by the increased throughput achieved by processing fewer vision tokens in the downstream layers. See Sec. A.3.3 for more details.

**Influence of hierarchical compression**   We conducted experiments at the same compression layer $\lambda$ while varying $\mathcal{W}$, as presented in Table 6. Across different compression layers $\lambda$, the results show a similar general trend: there is more pronounced degradation with a global compression strategy $w = 1$, likely due to the inadvertent destruction of spatial locality (Xu et al., 2022; Pan et al., 2022; Song et al., 2024; Norouzi et al., 2024).

We further evaluate the relative merits of our hierarchical compression in Table 7. We conduct additional experiments that perform compression directly on the full set of vision tokens (global-only), and only within fixed windows (local-only). Our results demonstrate that both variants underperform when compared to our strategy. This suggests that the efficacy of LightKV stems from the progressive expansion of the compression scope across stages, which balances local feature preservation with integration of global semantics.

Lastly, we summarize the FLOPs and KV cache memory usage for different inference configurations in Table 8, which shows that changing $\mathcal{W}$ has limited impact on aggregated FLOPs and memory under the same compression schedule.

Table 7: Performance comparison to global-only and local-only compression at 55% vision token retention.

| Strategy | Coco | MME | | NC | Pope | | Seed | VW | Avg % |
|---|---|---|---|---|---|---|---|---|---|
| | | C | P | | Acc | F1 | | | |
| **LLaVA-v1.5-13B** | | | | | | | | | |
| Global-only | 1.15 | 299.6 | 1530.2 | 1.08 | 0.87 | 0.85 | 0.39 | 0.55 | 93.92 |
| Local-only | 1.15 | 290.0 | 1529.9 | 1.08 | 0.87 | 0.85 | 0.38 | 0.56 | 93.55 |
| Ours | 1.15 | 302.1 | 1543.8 | 1.08 | 0.87 | 0.86 | 0.69 | 0.56 | **99.94** |
| **LLaVA-NeXT-13B** | | | | | | | | | |
| Global-only | 0.98 | 311.1 | 1549.8 | 0.85 | 0.87 | 0.86 | 0.34 | 0.60 | 91.31 |
| Local-only | 0.97 | 318.5 | 1543.8 | 0.85 | 0.87 | 0.86 | 0.34 | 0.60 | 91.44 |
| Ours | 0.96 | 326.1 | 1576.5 | 0.83 | 0.87 | 0.86 | 0.69 | 0.61 | **98.12** |

Table 8: Profiling results by varying compression layers $\Lambda$ and window sizes $\mathcal{W}$ on LLaVA 13B models.

| Method | $\Lambda$ | $\mathcal{W}$ | LLaVA-v1.5-13B | | LLaVA-NeXT-13B | |
|---|---|---|---|---|---|---|
| | | | FLOPs | Mem | FLOPs | Mem |
| Vanilla | - | - | 19.4 | 0.55 | 65.0 | 1.75 |
| LightKV | 15,23,31 | 4,2,1 | 12.6 | 0.37 | 37.3 | 1.05 |
| | | 6,4,2 | 12.6 | 0.37 | 37.3 | 1.05 |
| | 17,24,31 | 4,2,1 | 13.1 | 0.38 | 39.0 | 1.09 |
| | | 6,4,2 | 13.1 | 0.38 | 39.0 | 1.09 |

**Influence of compression layers**  We investigate the impact of varying layers for token compression, as illustrated in Figure 6 in the appendix. Trends between the compression layer and model performance reveal that compressing in the shallow layers has a more substantial impact on performance. This effect is particularly pronounced in VizWiz, where LVLMs must refrain from answering (*e.g.*, when the ground truth is "unanswerable"). Compression in the deeper layers yields performance nearly identical to the base LVLM models, but offers little reduction in memory usage.

Additional ablation studies, including bipartite vs. full pairwise matching for computing FD (Tables 12 and 13), similarity metrics (Table 16), and FastV under hyperparameter tuning (Table 18) are provided in Appendix Sec. A.3.2.

## 5   Conclusion

In this paper, we present LightKV, a novel *training-free* approach for optimizing KV cache storage for general-purpose LVLMs. It leverages *text-prompt-guided graph message passing and aggregation* to informatively compress vision tokens during the *prefill* stage of inference. Our method is designed to be: (i) memory-efficient: by progressively and dynamically compressing vision tokens through a hierarchical process; and (ii) compute-efficient: by employing window-based graph partitioning and bipartite matching to accelerate message aggregation. The experimental results demonstrate that our approach: (a) largely preserves the general-purpose performance of the base LVLM across multiple benchmarks, and (b) outperforms existing baselines in performance-efficiency trade-off.

**Limitations**  We acknowledge two limitations: (a) LightKV leverages a bipartite graph matching algorithm, which splits vision tokens into two disjoint sets, then finds low-FD pairings between nodes across the two sets. This limits the compression rate to a maximum of 50% per step, thus requiring multiple iterations to achieve higher overall reduction. (b) Furthermore, our method explicitly computes attention matrices for cross-modality guidance during a *small number* of compression steps, similar to prior approaches (Chen et al., 2024b; Liu et al., 2023a). These steps are less compatible with IO-efficient implementations such as FlashAttention (Dao et al., 2022), which do not expose the full attention matrix. However, layers where compression is not applied remain fully compatible with FlashAttention.

## Acknowledgments

We gratefully acknowledge the support of the NUS Artificial Intelligence Institute (NAII) through seed grant number NAII-SG-2025-027.

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

# A Appendix

## A.1 Summary of notations

Table 9 provides an overview of the notations used in this paper.

Table 9: Summary of notations.

| Notation | Definition |
|---|---|
| $L$ | Total number of decoder layers in the LVLM |
| $s$ | Number of compression iterations, with $s < L$ |
| $\lambda_i$ | Decoder layer where the $i$-th compression step is performed |
| $\Lambda = [\lambda_1, \ldots, \lambda_s]$ | Schedule of compression layers |
| $w_i$ | Number of window partitions *along each axis* used at stage $i$ |
| $w_i^2$ | Number of window partitions used at stage $i$ |
| $\mathcal{W} = [w_1, \ldots, w_s]$ | Schedule of window partitions, with $w_i > w_{i+1}$ |
| $\rho_i$ | Compression ratio applied at stage $i$ |
| $\mathcal{P} = [\rho_1, \ldots, \rho_s]$ | Schedule of compression rates |
| $X_v$ | Sequence / set of vision tokens |
| $v$ | Number of vision tokens |
| $\omega$ | Index for an individual window |
| $v_\omega = \frac{v}{w \times w}$ | Number of vision tokens in window $\omega$ |
| $\mathbf{x}$ | Embedding of a vision token / graph node |
| $\mathcal{X} = \{\mathbf{x} \mid \mathbf{x} \in X_v\}$ | Set of graph nodes formed from vision tokens |
| $\mathcal{X}_A, \mathcal{X}_B$ | Two near-equal subsets of nodes in the bipartite graph |
| $\mathcal{E} = \mathcal{X}_A \times \mathcal{X}_B$ | Edges of the bipartite graph |
| $M \in \{0,1\}^{\lvert \mathcal{X}_A \rvert \times \lvert \mathcal{X}_B \rvert}$ | Rectangular adjacency matrix for selected bipartite edges |
| $\mathrm{FD}(\alpha, \beta)$ | Feature divergence between two nodes with indices $\alpha$ and $\beta$ |
| $\mathcal{T}_\rho$ | Set of the $\lfloor \rho v_\omega / 2 \rfloor$ selected token pairs with smallest FD values |
| $\mathcal{X}_R$ | Unconnected nodes left unchanged after edge selection |
| $A \in \mathbb{R}^{H \times (p+v) \times (p+v)}$ | Multi-head attention matrix during prefill |
| $H$ | Number of attention heads |
| $p$ | Number of prompt tokens |
| $\mathcal{J}$ | Index set of prompt tokens |
| $\xi_i$ | Prompt-guidance weight / accumulated attention for vision token $i$ |
| $\xi_A, \xi_B$ | Prompt-guidance weights corresponding to $X_A$ and $X_B$ |

## A.2 Method

### A.2.1 Method overview

As illustrated in Fig. 5, we insert graph message passing-based compression between two selected decoder layers in the LVLM, simultaneously reducing KV cache size and the number of vision tokens processed by downstream layers. Compression is performed three times to achieve the overall compression ratio.

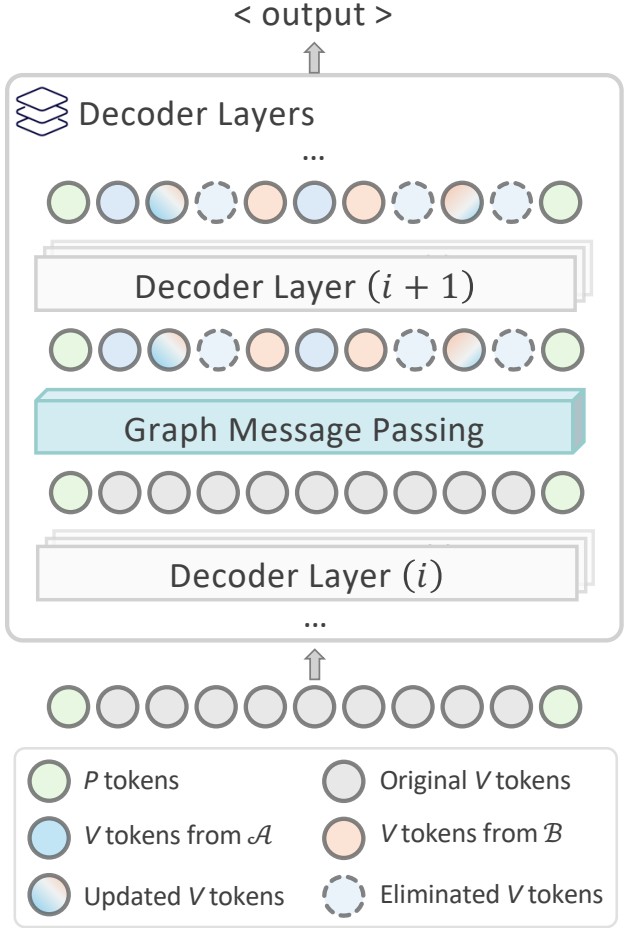

Figure 5: LightKV dynamically compresses vision tokens between two consecutive LVLM decoder layers. The key and value tokens are compressed simultaneously for later layers, reducing the memory used by KV cache.

### A.2.2 Adjacency matrix

In Sec. 3.2, we defined for our bipartite graph with edges $M \in \{0,1\}^{|\mathcal{X}_\mathcal{A}| \times |\mathcal{X}_\mathcal{B}|}$, whose rows correspond to nodes in $\mathcal{X}_\mathcal{A}$ and columns to nodes in $\mathcal{X}_\mathcal{B}$. However, as the two subsets need not contain the same number of nodes, $M$ is generally rectangular. Conventionally, for a standard graph, the adjacency matrix is square with side length equal to the total number of nodes. The analogous square adjacency matrix for our bipartite graph is:

$$\begin{pmatrix} 0 & M \\ M^\top & 0 \end{pmatrix}, \tag{9}$$

where the upper-left and lower-right blocks are zero by definition. Throughout our paper, we work directly with $M$, as this rectangular form is sufficient for message passing between the two partitions.

### A.3 Additional results

### A.3.1 Additional backbones

**Qwen2.5-VL** We also evaluated LightKV on Qwen2.5-VL-7B-Instruct (Bai et al., 2025) across multiple compression ratios. The results in Table 10 demonstrate that LightKV yields substantial improvements compared to baseline approaches, preserving accuracy more effectively and delivering stronger overall performance under compression. Notably, as presented in Table 11, at more aggressive compression ratios, LightKV still delivers near-identical performance to the vanilla model.

Table 10: Results of LightKV on Qwen2.5-VL-7B-Instruct model at 55% vision token retention in the KV cache. **Avg %** denotes the average of all performance metrics normalized against the Vanilla model. Methods are then sorted by average score. "NC" and "VW" denote NoCaps and VizWiz, respectively.

| Method | Coco | GQA | MME | | NC | Pope | | Seed | VW | Avg % |
|--------|------|-----|-----|---|----|------|---|------|-----|-------|
| | | | C | P | | Acc | F1 | | | |
| Vanilla | 0.319 | 0.604 | 638.21 | 1695.25 | 0.372 | 0.875 | 0.862 | 0.790 | 0.704 | 100.00 |
| FastV | 0.339 | 0.587 | 625.35 | 1687.78 | 0.386 | 0.869 | 0.853 | 0.744 | 0.698 | 98.77 |
| ToMe | 0.329 | 0.591 | 640.71 | 1687.75 | 0.425 | 0.862 | 0.782 | 0.782 | 0.683 | 100.04 |
| PiToMe | 0.389 | 0.584 | 624.64 | 1671.09 | 0.433 | 0.860 | 0.842 | 0.774 | 0.691 | 100.24 |
| ToFu | 0.383 | 0.587 | 657.86 | 1683.05 | 0.418 | 0.857 | 0.839 | 0.788 | 0.696 | 100.75 |
| LightKV | 0.389 | 0.591 | 647.50 | 1706.38 | 0.435 | 0.863 | 0.846 | 0.780 | 0.694 | **101.37** |

Table 11: Results of LightKV on Qwen2.5-VL-7B-Instruct model at various retention rates of vision tokens in the KV cache. **Avg %** denotes the average of all performance metrics normalized against the Vanilla model. "NC" and "VW" denote NoCaps and VizWiz, respectively.

| Rate | Coco | GQA | MME | | NC | Pope | | Seed | VW | Avg % |
|------|------|-----|-----|---|----|------|---|------|-----|-------|
| | | | C | P | | Acc | F1 | | | |
| Vanilla | 0.319 | 0.604 | 638.21 | 1695.25 | 0.372 | 0.875 | 0.862 | 0.790 | 0.704 | 100.00 |
| 55% | 0.389 | 0.591 | 647.50 | 1706.38 | 0.435 | 0.863 | 0.846 | 0.780 | 0.694 | 101.37 |
| 40% | 0.370 | 0.586 | 611.78 | 1632.64 | 0.450 | 0.851 | 0.830 | 0.754 | 0.666 | 101.01 |
| 30% | 0.361 | 0.581 | 588.93 | 1574.34 | 0.455 | 0.833 | 0.806 | 0.732 | 0.670 | 98.89 |
| 20% | 0.356 | 0.569 | 591.78 | 1612.83 | 0.458 | 0.835 | 0.809 | 0.730 | 0.667 | 99.24 |

### A.3.2 Additional ablation studies

**Bipartite vs. full pairwise matching** We provide additional ablation studies to analyze the design choice of bipartite matching compared to full pairwise matching. We evaluate both approaches from two perspectives: (1) downstream task performance and (2) computational efficiency.

While bipartite matching does not guarantee globally optimal pair assignments, we empirically observe that its impact on downstream performance is marginal. The results are shown in Table 12: across all benchmarks, bipartite matching achieves comparable performance to full pairwise matching.

We hypothesize that this behavior is due to our multi-stage compression strategy. Although globally optimal pairs may not be matched in early stages (*e.g.*, when tokens fall into the same partition), these tokens are likely to be reassigned into different partitions in later stages, where they can then be matched and merged. This progressively mitigates the sub-optimality introduced by bipartite partitioning.

We further compare the computational cost of the two matching strategies. As derived in Sec. 3.2.1, bipartite matching reduces the number of similarity comparisons from $\mathcal{O}(v_w^2/2)$ to $\mathcal{O}(v_w^2/4)$, effectively halving the pairwise operations. In practice, however, we observe an even larger gap in runtime cost. As shown in Table 13, full pairwise matching incurs approximately $4\times$ higher FLOPs than bipartite matching across different numbers of vision tokens. This is due to additional overhead in computing and maintaining the full similarity matrix. The increased computation also translates to higher memory (VRAM) usage.

Overall, bipartite matching provides a favorable trade-off between performance and efficiency.

**Influence of window schedule** Table 14 studies the effect of window schedule $\mathcal{W}$, which is closely related to the number of vision tokens used by the LVLM. A larger initial window size is appropriate when the model encodes images at high resolution, e.g., LLaVA-NeXT encodes an image into 2,144 tokens. In contrast,

Table 12: Performance comparison between bipartite matching and full pairwise matching on LLaVA-v1.5 when retaining 55% of vision tokens.

| Method | MME | | Pope | | Avg % |
|---|---|---|---|---|---|
| | C | P | Acc. | F1 | |
| **LLaVA-v1.5-13B** | | | | | |
| Bipartite | 302.1 | 1543.8 | 0.87 | 0.86 | 100.75 |
| Full Pairwise | 298.9 | 1532.0 | 0.87 | 0.86 | 100.31 |
| **LLaVA-v1.5-7B** | | | | | |
| Bipartite | 357.5 | 1519.8 | 0.87 | 0.86 | 100.30 |
| Full Pairwise | 371.0 | 1522.2 | 0.86 | 0.85 | 100.71 |

Table 13: FLOPs comparison of bipartite and full pairwise matching across vision-token counts.

| Method | # Vision tokens | | | |
|---|---|---|---|---|
| | 512 | 1024 | 2048 | 4096 |
| Bipartite | 0.134 | 0.538 | 2.151 | 8.602 |
| Full Pairwise | 0.538 | 2.151 | 8.603 | 34.410 |

Table 14: Performance comparison across various combinations of $\mathcal{W}$ on LLaVA-13B models at 55% vision token retention. "NC" and "VW" denote NoCaps and VizWiz, respectively.

| Method | Coco | MME | | NC | Pope | | SQA | Seed | VW | **Avg %** |
|---|---|---|---|---|---|---|---|---|---|---|
| | | C | P | | Acc | F1 | | | | |
| **LLaVA-v1.5-13B** | | | | | | | | | | |
| Vanilla | 1.16 | 295.36 | 1532.0 | 1.09 | 0.87 | 0.86 | 0.73 | 0.69 | 0.57 | 100.00 |
| LightKV $\mathcal{W}$=[4,2,1] | 1.15 | 302.14 | 1543.8 | 1.08 | 0.87 | 0.86 | 0.72 | 0.69 | 0.56 | 99.80 |
| LightKV $\mathcal{W}$=[6,4,2] | 1.14 | 301.79 | 1541.1 | 1.08 | 0.87 | 0.86 | 0.72 | 0.69 | 0.56 | 99.67 |
| **LLaVA-NeXT-13B** | | | | | | | | | | |
| Vanilla | 1.02 | 318.93 | 1575.1 | 0.88 | 0.88 | 0.86 | 0.73 | 0.69 | 0.64 | 100.00 |
| LightKV $\mathcal{W}$=[4,2,1] | 0.96 | 311.43 | 1576.3 | 0.83 | 0.87 | 0.86 | 0.59 | 0.69 | 0.61 | 95.68 |
| LightKV $\mathcal{W}$=[6,4,2] | 0.96 | 326.07 | 1576.5 | 0.83 | 0.87 | 0.86 | 0.59 | 0.69 | 0.61 | 96.20 |

a smaller value of $w$ is more favorable when there are fewer vision tokens, e.g., LLaVA-v1.5, which uses 576 vision tokens per image. In our experiments, we used $\mathcal{W} = [6, 4, 2]$ for LLaVA-NeXT and $\mathcal{W} = [4, 2, 1]$ for LLaVA-v1.5. We found that using a large window size with fewer vision tokens overly restricts token matching, often resulting in mismatches.

**Influence of compression layers**   We investigate the impact of varying layers for token compression, as illustrated in Fig. 6. Trends between the compression layer and model performance reveal that compressing in the shallow layers has a more substantial impact on performance. This effect is particularly pronounced in VizWiz, where LVLMs must refrain from answering (*e.g.*, when the ground truth is "unanswerable"). Compression in the deeper layers yields performance nearly identical to the base LVLM models, but offers little reduction in memory usage.

**Overall robustness to compression schedule**   The compression schedule in our method is designed heuristically rather than learned from data. This choice is intentional: our goal is to provide a training-free,

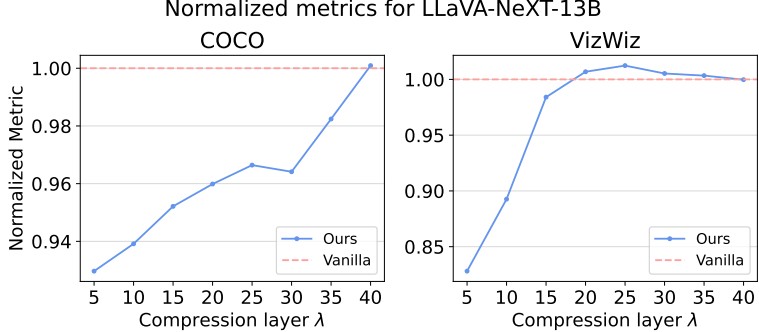

Figure 6: Performance comparison on LLaVA-NeXT-13B under different compression layer choices $\lambda$.

Table 15: Performance comparison across different schedules by varying $\Lambda$ and $\mathcal{W}$ at 55% vision token retention. Our method is robust to the layer and window schedules.

| $\Lambda$ | $\mathcal{W}$ | Coco | MME | | NC | Pope | | Seed | VW | Avg % |
| | | | C | P | | Acc | F1 | | | |
|---|---|---|---|---|---|---|---|---|---|---|
| | | | | **LLaVA-v1.5-13B** | | | | | | |
| $15, 23, 31$ | $4, 2, 1$ | 1.15 | 302.1 | 1543.8 | 1.08 | 0.87 | 0.86 | 0.69 | 0.56 | **99.94** |
| $15, 23, 31$ | $6, 4, 2$ | 1.15 | 295.5 | 1530.8 | 1.08 | 0.88 | 0.86 | 0.69 | 0.57 | 99.92 |
| $9, 19, 29$ | $4, 2, 1$ | 1.17 | 263.6 | 1518.9 | 1.07 | 0.87 | 0.86 | 0.69 | 0.52 | 97.32 |
| $9, 19, 29$ | $6, 4, 2$ | 1.17 | 263.7 | 1518.9 | 1.07 | 0.87 | 0.86 | 0.68 | 0.53 | 97.38 |
| | | | | **LLaVA-v1.5-7B** | | | | | | |
| $12, 18, 24$ | $6, 4, 2$ | 1.11 | 357.5 | 1519.8 | 1.03 | 0.87 | 0.86 | 0.66 | 0.53 | **99.79** |
| $12, 18, 24$ | $4, 2, 1$ | 1.11 | 354.3 | 1512.6 | 1.04 | 0.87 | 0.85 | 0.67 | 0.53 | 99.66 |

plug-and-play solution that can be readily applied to arbitrary LVLMs without incurring additional training cost or requiring a learned policy for schedule selection. To promote generalization and avoid task-specific bias, we determine the schedule parameters using a subset of benchmarks (COCO and MME), and then fix them across all remaining tasks. This protocol reduces the risk of implicitly overfitting the schedule to any particular evaluation setting. While learning an adaptive scheduler is an interesting direction for future work, our results suggest that such complexity may not be necessary for strong performance. As shown in Table 15, performance remains stable across a range of $\Lambda$ and $\mathcal{W}$ configurations, indicating that the method is robust to the choice of schedule.

**Influence of similarity metrics** We evaluate the impact of different similarity metrics used in the token-pairing process of LightKV, as first described in Sec. 3.2.1. Table 16 compares the results of cosine similarity to Euclidean distance and L2-Squared distance. Overall, cosine similarity consistently achieves the best and most stable performance across benchmarks. In contrast, Euclidean and L2-Squared distances lead to noticeable degradation, particularly on tasks such as SeedBench and MME. Based on these observations, we adopt cosine similarity as the default metric for token pairing in LightKV.

### A.3.3 Additional latency profiles

We evaluate model responsiveness using two latency metrics: time-to-first-token (TTFT) and generation latency for 100 tokens. As shown in Table 17, TTFT highlights the overhead of the prefilling stage and directly reflects user-perceived responsiveness, while generation latency characterizes decoding efficiency. Together, these results provide a comprehensive view of both initial response delay and sustained throughput.

Table 16: Performance comparison of using cosine similarity, Euclidean distance and L2-Squared distance at 55% vision token retention.

| Metric | Coco | MME | | NC | Pope | | Seed | VW | Avg % |
|--------|------|-----|---|----|------|---|------|----|----|
| | | C | P | | Acc | F1 | | | |
| **LLaVA-v1.5-13B** | | | | | | | | | |
| Euclidean | 1.15 | 297.5 | 1529.2 | 1.07 | 0.87 | 0.86 | 0.39 | 0.55 | 93.85 |
| L2-Squared | 1.15 | 292.5 | 1530.0 | 1.08 | 0.87 | 0.86 | 0.39 | 0.55 | 93.77 |
| Cosine | 1.15 | 302.1 | 1543.8 | 1.08 | 0.87 | 0.86 | 0.69 | 0.56 | **99.94** |
| **LLaVA-NeXT-13B** | | | | | | | | | |
| Euclidean | 0.96 | 316.0 | 1553.9 | 0.84 | 0.87 | 0.86 | 0.34 | 0.59 | 90.96 |
| L2-Squared | 0.98 | 308.5 | 1554.7 | 0.84 | 0.87 | 0.86 | 0.35 | 0.60 | 91.29 |
| Cosine | 0.96 | 326.1 | 1576.5 | 0.83 | 0.87 | 0.86 | 0.69 | 0.61 | **98.12** |

Table 17: Latency comparison across LLaVA models. TTFT = Time to First Token. Gen latency = latency for generating 100 tokens. Lower is better.

| Method | TTFT (ms) | Gen latency (s) | TTFT (ms) | Gen latency (s) |
|--------|-----------|-----------------|-----------|-----------------|
| | **LLaVA-v1.5-13B** | | **LLaVA-v1.5-7B** | |
| Vanilla | $130 \pm 0.393$ | $2.89 \pm 0.004$ | $78 \pm 0.827$ | $2.09 \pm 0.008$ |
| FastV | $85 \pm 0.450$ | $2.59 \pm 0.003$ | $52 \pm 0.316$ | $1.85 \pm 0.004$ |
| PiToMe | $93 \pm 0.768$ | $2.79 \pm 0.017$ | $58 \pm 1.543$ | $2.14 \pm 0.010$ |
| ToFu | $94 \pm 1.152$ | $2.83 \pm 0.009$ | $58 \pm 0.428$ | $2.14 \pm 0.004$ |
| ToMe (P) | $94 \pm 0.278$ | $2.77 \pm 0.003$ | $58 \pm 0.647$ | $2.13 \pm 0.003$ |
| LightKV | $98 \pm 1.045$ | $2.80 \pm 0.004$ | $65 \pm 3.054$ | $2.11 \pm 0.006$ |
| | **LLaVA-NeXT-13B** | | **LLaVA-NeXT-7B** | |
| Vanilla | $656 \pm 1.479$ | $3.79 \pm 0.013$ | $397 \pm 1.337$ | $2.43 \pm 0.005$ |
| FastV | $321 \pm 2.142$ | $3.03 \pm 0.003$ | $197 \pm 1.207$ | $2.03 \pm 0.004$ |
| PiToMe | $396 \pm 1.109$ | $3.30 \pm 0.004$ | $247 \pm 0.633$ | $2.34 \pm 0.005$ |
| ToFu | $394 \pm 0.968$ | $3.30 \pm 0.003$ | $245 \pm 0.757$ | $2.35 \pm 0.005$ |
| ToMe (P) | $394 \pm 1.280$ | $3.30 \pm 0.004$ | $245 \pm 0.722$ | $2.32 \pm 0.002$ |
| LightKV | $383 \pm 0.878$ | $3.29 \pm 0.002$ | $259 \pm 0.643$ | $2.30 \pm 0.004$ |

### A.3.4 Performance comparison to FastV

We provide additional experiments to ensure a fair comparison with the FastV baseline. In our main experiments, we followed the default FastV configuration as described in its original implementation, where pruning is performed at an early transformer layer (specifically, layer index $K = 2$). While this is a key design choice of FastV, it may not fully reflect its best achievable performance under different configurations.

To account for this, we conduct a more comprehensive evaluation by varying the pruning layer $K \in \{1, 2, 4, 8\}$. To ensure a controlled comparison, we adjust the retention ratio $R$ such that all variants maintain the same overall retention rate of vision tokens in the KV cache (55%).

The results are summarized in Table 18. Across both LLaVA-v1.5-7B and LLaVA-NeXT-7B, LightKV consistently achieves competitive or superior performance compared to FastV under different choices of $K$. Notably, while certain configurations of FastV (e.g., larger $K$) can partially recover performance, they still do not consistently surpass LightKV under the same compression budget.

### A.3.5 Visualization

We provide visualization cases for vision token compression of Coco images in Fig. 7 for a 3-stage compression on LLaVA-v1.5-13B, reducing the number of tokens from $576 \rightarrow 288 \rightarrow 145 \rightarrow 77$. Unlike conventional vision encoders, vision tokens in LVLMs incorporate prompt information. As a result, visually similar patches may

Table 18: Performance comparison between LightKV and FastV under different pruning layers $K$ at 55% vision token retention.

| Method | | Coco | MME | | NC | Pope | | Seed | VW | **Avg %** |
|---|---|---|---|---|---|---|---|---|---|---|
| | | | C | P | | Acc | F1 | | | |
| **LLaVA-v1.5-7B** | | | | | | | | | | |
| FastV | $K=1$ | 1.08 | 337.8 | 1469.7 | 1.02 | 0.83 | 0.81 | 0.57 | 0.54 | 95.45 |
| | $K=2$ | 1.10 | 351.1 | 1513.7 | 1.04 | 0.85 | 0.83 | 0.66 | 0.54 | 99.03 |
| | $K=4$ | 1.10 | 339.2 | 1500.0 | 1.04 | 0.84 | 0.81 | 0.66 | 0.54 | 98.06 |
| | $K=8$ | 1.10 | 371.7 | 1502.0 | 1.02 | 0.85 | 0.84 | 0.65 | 0.53 | 99.13 |
| LightKV | | 1.11 | 357.5 | 1519.8 | 1.03 | 0.87 | 0.86 | 0.66 | 0.53 | **99.79** |
| **LLaVA-NeXT-7B** | | | | | | | | | | |
| FastV | $K=1$ | 0.96 | 326.0 | 1495.2 | 0.85 | 0.86 | 0.84 | 0.68 | 0.60 | 97.87 |
| | $K=2$ | 0.88 | 265.4 | 1341.3 | 0.78 | 0.81 | 0.77 | 0.69 | 0.58 | 90.37 |
| | $K=4$ | 0.98 | 298.9 | 1504.5 | 0.86 | 0.87 | 0.85 | 0.68 | 0.59 | 97.38 |
| | $K=8$ | 0.96 | 293.2 | 1505.7 | 0.84 | 0.87 | 0.84 | 0.67 | 0.60 | 96.52 |
| LightKV | | 0.98 | 338.6 | 1517.3 | 0.83 | 0.88 | 0.86 | 0.69 | 0.58 | **98.85** |

differ significantly in the embedding space, making it plausible to aggregate non-adjacent patches. To this end, our intra-window strategy imposes constraints on this aggregation process to maintain spatial coherence during compression.

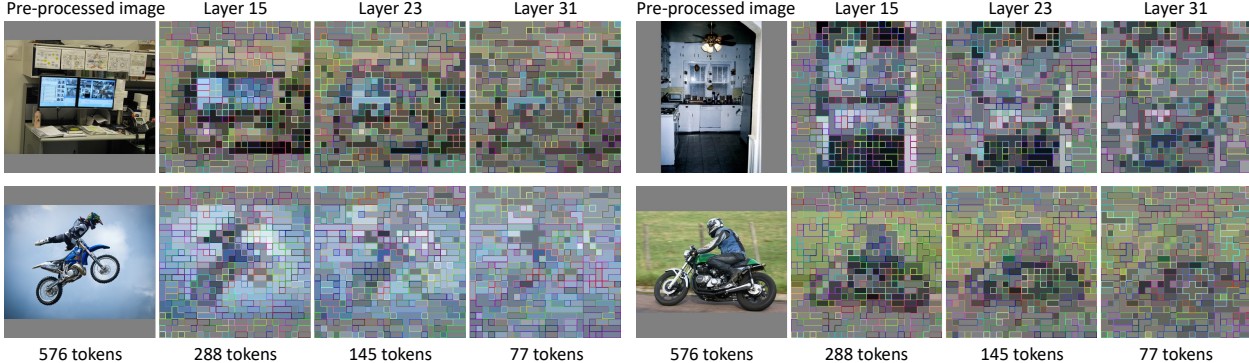

Figure 7: Visualization of a 3-stage vision token compression, halving tokens at each stage and achieving 55% vision token retention in the KV cache. Distant patches may be compressed into a single token.

