# OpenReview forum: "Make Your LVLM KV Cache More Lightweight"
_TMLR — Accepted by TMLR_

### Review · Reviewer_GidE · 2026-03-11

**Summary Of Contributions:**

This paper proposes LightKV, a training-free method for reducing the memory footprint of the KV cache in Large Vision-Language Models (LVLMs) during inference. The key idea is to compress vision tokens during the prefill stage by exploiting redundancy among image tokens while preserving tokens that are most relevant to the text prompt.

The method operates inside the transformer decoder layers and performs hierarchical token compression. Vision tokens are partitioned into spatial windows and mapped to nodes in a bipartite graph. Tokens with small feature divergence (FD) are paired and aggregated via a prompt-guided message passing mechanism that uses cross-modal attention from text tokens to weight token importance. After aggregation, redundant tokens are removed and the remaining tokens are propagated to subsequent layers.

The authors evaluate LightKV across eight LVLMs and eight benchmarks, showing that compressing roughly 50% of vision tokens can substantially reduce KV cache size and computational cost while preserving performance across a variety of multimodal tasks.

Overall, the work addresses a practical efficiency problem in LVLM inference and proposes a plug-and-play method that can be applied to multiple model architectures without retraining.

**Audience:**

Yes

**Audience Explanation:**

The efficiency of large multimodal models is an increasingly important topic. LVLMs typically process hundreds or thousands of visual tokens, which leads to substantial memory and computational overhead during inference. Methods that reduce these costs without retraining models are therefore highly relevant to both researchers and practitioners.

LightKV addresses this problem by proposing a **training-free token compression method** that can be applied to multiple LVLM architectures. The approach combines token merging ideas with cross-modal attention to guide compression, which may inspire further work on efficient multimodal inference.

Because of its practical focus and compatibility with existing models, the paper is likely to interest researchers working on:

- efficient transformer inference
- multimodal large models
- token pruning and merging methods
- systems optimization for generative AI

**Broader Impact Concerns:**

I do not see major negative societal concerns specific to this work. The method focuses on improving the efficiency of multimodal model inference and may help reduce computational and energy costs associated with deploying large vision-language models.

As with most work on large generative models, improved efficiency may indirectly enable wider deployment of such systems, which raises broader concerns related to misuse, bias, or misinformation. However, these concerns are not unique to this paper and are already widely discussed in the literature.

Overall, the work appears to have primarily positive impact by enabling more efficient use of computational resources.

**Claims And Evidence:**

Yes

**Claims Explanation:**

The paper presents extensive experiments across several open-source LVLMs and benchmarks, reporting both **efficiency metrics** (FLOPs, memory usage, latency) and **task performance**. The results suggest that LightKV consistently reduces KV memory usage and computational cost while preserving performance across a variety of multimodal tasks.

The evaluation spans multiple models, datasets, and compression rates, which strengthens the empirical support for the claims. In addition, the comparisons against several existing token compression or pruning methods provide useful context for evaluating the method.

However, some aspects of the algorithmic description could be clarified to improve the technical presentation.

**1. Graph partitioning**

It is not clearly specified how the vision tokens are split into the two sets $X_A$ and $X_B$. It appears that the split may be arbitrary or random, but this is not explicitly stated. If the split is random, it would be useful to understand how far this may deviate from an optimal pairing strategy and whether the results are sensitive to this choice.

**2. Feature divergence metric**

The feature divergence metric corresponds to cosine distance. While this is a reasonable and commonly used similarity metric, the paper could clarify whether alternative metrics were considered.

**3. Definition of $\mathcal{T}_{\rho}$**

The set $\mathcal{T}_{\rho}$ is introduced when defining the adjacency matrix, but it is not immediately clear how it is constructed. From the description it appears to correspond to selecting the pairs with the smallest FD values according to the compression ratio, but this could be defined more explicitly.

**4. Bipartite matching description**

The paper describes the token pairing step as bipartite matching. However, the algorithm appears to select the smallest FD edges without enforcing a one-to-one matching constraint. This allows multiple edges to share the same node, meaning the procedure is not a strict bipartite matching but rather a many-to-one aggregation scheme. Clarifying this distinction would improve the methodological description.

**5. Compression schedule notation**

The notation for the compression schedule $(\Lambda, P, W)$ is introduced somewhat abruptly and inconsistently. In particular, the paper alternates between $P$ and $\rho$ when referring to compression ratios. Providing clearer definitions when these variables are first introduced would improve readability.

These issues mainly concern clarity of presentation rather than correctness and could likely be addressed with minor revisions.

**Requested Changes:**

I recommend acceptance, but I suggest the authors address the following issues to improve clarity:

1. Clarify how the token partition into $X_A$ and $X_B$ is performed and whether the method is sensitive to this choice.
2. Provide a clearer definition of $\mathcal{T}_{\rho}$ when it is first introduced.
3. Clarify whether the pairing step enforces a one-to-one matching constraint or whether it allows many-to-one aggregation.
4. Improve notation consistency, particularly for the compression schedule parameters $(\Lambda, P, W)$ and the compression ratios $\rho$.
5. Briefly discuss whether alternative similarity metrics or pairing strategies were considered.

These revisions would improve the clarity and reproducibility of the method.

---

> ### Author Response · Authors · 2026-04-04
> **Response to reviewer GidE (part 1)**
>
> We thank the reviewer for the valuable feedback on our manuscript. We respond to the concerns raised below.
>
> - **[Q1] Clarify how the token partition into $X_A$ and $X_B$ is performed and whether the method is sensitive to this choice.**
>
>     We thank the reviewer for raising this comment. Our approach follows the bipartite matching strategy, as seen in prior works such as ToMe [1], and partitions the tokens into the two sets using an alternating index (odd for $\mathcal{X}_A$ and even for $\mathcal{X}_B$). This tends to split neighbouring nodes across the two sets, allowing them to be merged. Furthermore, the use of bipartite matching preserves the core benefit of similarity-based full pairwise matching while avoiding the heavy computation required.
>
>     In addition, Tables 1 and 2 below compare the performance between bipartite matching and full matching, with significant drops in performance across multiple benchmarks when using full pairwise matching.
>
>     **Table 1**: Performance comparison of bipartite matching to full pairwise matching on LLaVA-v1.5-13B.
>
>     | **Method** | **MME-C** | **MME-P** | **Pope-Acc** | **Pope-F1** | **Avg** |
>     | --- | --- | --- | --- | --- | --- |
>     | Bipartite | 302.1 | 1543.8 | 0.87 | 0.86 | 100.75 |
>     | Full pairwise | 298.9 | 1532.0 | 0.87 | 0.86 | 100.30 |
>
>     **Table 2**: Performance comparison of bipartite matching to full pairwise matching on LLaVA-v1.5-7B.
>
>     | **Method** | **MME-C** | **MME-P** | **Pope-Acc** | **Pope-F1** | **Avg** |
>     | --- | --- | --- | --- | --- | --- |
>     | Bipartite | 357.5 | 1519.8 | 0.87 | 0.86 | 100.30 |
>     | Full pairwise | 371.0 | 1522.2 | 0.86 | 0.85 | 100.71 |
>
>     Overall, we believe our method achieves a favorable trade-off between model performance and computational efficiency.
>
>     **References:**
>
>     [1] Bolya, Daniel, et al. "Token merging: Your vit but faster." *ICLR* 2023.
>
> - **[Q2] Provide a clearer definition of $\mathcal{T}_\rho$ when it is first introduced.**
>
>     We sincerely thank the reviewer for pointing this out. $\mathcal{T}\_{\rho}$ denotes the set of the $(\rho v / 2)$ token pairs $(\alpha,\beta)$ with the smallest $FD(\alpha,\beta)$ values among all pairs with $\alpha \in \mathcal{X}\_\mathcal{A}$ and $\beta \in \mathcal{X}\_\mathcal{B}$. Concretely, we compute $FD(\alpha,\beta)$ for every such pair, rank all pairs in ascending order of $FD$, and select the top $(\rho v / 2)$ pairs to form $\mathcal{T}\_{\rho}$. We have revised the manuscript accordingly to make this definition explicit.
>
>
> - **[Q3] Clarify whether the pairing step enforces a one-to-one matching constraint or whether it allows many-to-one aggregation.**
>
>     We apologize for the confusion and sincerely thank the reviewer for the constructive comment. We use a many-to-one aggregation relationship. Firstly, the unconnected nodes ($\mathcal{X}\_R=\{\mathbf{x}\_r|(r,\beta)\notin \mathcal{T}\_\rho\}$) are excluded from pairing and are therefore not aggregated. Secondly, pairing does not impose a one-to-one constraint. Each remaining token in $\mathcal{X}\_\mathcal{A}\setminus \mathcal{X}\_R$ is assigned a token in $\mathcal{X}\_\mathcal{B}$, i.e., $\forall \alpha \in \mathcal{X}\_A \setminus \mathcal{X}\_R,\ \exists \beta \in \mathcal{X}\_B \text{ s.t. } (\alpha,\beta)\in \mathcal{T}\_\rho$. Although, the reverse may not hold: a token in $\mathcal{X}\_{\mathcal{B}}$ may be paired with multiple tokens from $\mathcal{X}\_{\mathcal{A}}$. Thus, pairing allows many-to-one aggregation, although it does not require it. This behavior is illustrated in Step 2 of Figure 2. After the message passing process, nodes in $\mathcal{X}\_{\mathcal{A}}$ are eliminated.

---

> ### Author Response · Authors · 2026-04-04
> **Response to reviewer GidE (part 2)**
>
> - **[Q4] Improve notation consistency, particularly for the compression schedule parameters and the compression ratios .**
>
>     We sincerely appreciate this thoughtful suggestion. We have updated our manuscript with the following changes to ensure consistency in the notations. We have made the style of the notations, such as calligraphic fonts, consistent with their definitions.
>
>     For better clarity, we included a table of the notations used in Table 9 of our updated manuscript, and summarized the compression schedule parameters in Table 4 below. These notations are consistent with our definitions in the original manuscript (other than the changes raised above).
>
>     **Table 4**: Summary of schedule parameters.
>
>     | **Notation** | **Definition** |
>     | --- | --- |
>     | $L$ | Total number of decoder layers in the LVLM |
>     | $s$ | Number of compression iterations, with $s < L$ |
>     | $\lambda_i$ | Decoder layer where the $i$-th compression step is performed |
>     | $\Lambda=[\lambda_1, \ldots, \lambda_s]$ | Schedule of compression layers |
>     | $w_i$ | Number of window partitions *per axis* at stage $i$ |
>     | $w_i^2$ | Number of window partitions used at stage $i$ |
>     | $\mathcal{W}=[w_1, \ldots, w_s]$ | Schedule of window partitions |
>     | $\rho_i$ | Compression ratio applied at stage $i$ |
>     | $\mathcal{P}=[\rho_1, \ldots, \rho_s]$ | Schedule of compression rates |
>
> - **[Q5] Briefly discuss whether alternative similarity metrics or pairing strategies were considered.**
>
>     We thank the reviewer for this insightful suggestion. In response, Tables 5 and 6  below present additional experiments using alternative similarity metrics, namely, Euclidean distance and L2-squared distance. Overall, cosine similarity yields the most stable performance across the evaluated benchmarks. Therefore, we employed cosine similarity in the pairing process in LightKV.
>
>     **Table 5**: Performance comparison of using cosine similarity, Euclidean distance, and L2-squared metrics on LLaVA-v1.5-13B.
>
>     | **Metric** | **Coco** | **MME-C** | **MME-P** | **NoCaps** | **Pope-Acc** | **Pope-F1** | **Seed** | **VizWiz** | **Avg** |
>     | --- | --- | --- | --- | --- | --- | --- | --- | --- | --- |
>     | Cosine | 1.15 | 302.1 | 1543.8 | 1.08 | 0.87 | 0.86 | 0.69 | 0.56 | **99.94** |
>     | Euclidean | 1.15 | 297.5 | 1529.2 | 1.07 | 0.87 | 0.86 | 0.39 | 0.55 | 93.85 |
>     | L2-Sq | 1.15 | 292.5 | 1530.0 | 1.08 | 0.87 | 0.86 | 0.39 | 0.55 | 93.77 |
>
>     **Table 6**: Performance comparison of using cosine similarity, Euclidean distance and L2-squared metrics on LLaVA-NeXT-13B.
>
>     | **Metric** | **Coco** | **MME-C** | **MME-P** | **NoCaps** | **Pope-Acc** | **Pope-F1** | **Seed** | **VizWiz** | **Avg** |
>     | --- | --- | --- | --- | --- | --- | --- | --- | --- | --- |
>     | Cosine | 0.96 | 326.1 | 1576.5 | 0.83 | 0.87 | 0.86 | 0.69 | 0.61 | **98.12** |
>     | Euclidean | 0.96 | 316.0 | 1553.9 | 0.84 | 0.87 | 0.86 | 0.34 | 0.59 | 90.96 |
>     | L2-Sq | 0.98 | 308.5 | 1554.7 | 0.84 | 0.87 | 0.86 | 0.35 | 0.60 | 91.29 |

---

> > ### Comment · Reviewer_GidE · 2026-04-04
> >
> > Thank you for the clear and thorough rebuttal. The clarifications on the token partitioning, pairing mechanism, and notation significantly improved the readability of the method. I also appreciate the additional experiments on alternative similarity metrics.
> >
> > Overall, the paper presents a solid and practical contribution to efficient LVLM inference, with strong empirical validation across models and benchmarks. I have no further major concerns.

---

### Review · Reviewer_cQ9i · 2026-03-15

**Summary Of Contributions:**

This paper studies KV-cache reduction for large vision-language models during the prefill stage, where vision tokens dominate memory usage. The proposed method, LightKV, compresses vision tokens without retraining by combining locality-preserving hierarchical windowing with bipartite graph message passing, and uses prompt-to-vision attention as a guidance signal when aggregating tokens. The paper evaluates the method on 8 LVLMs and 8 benchmarks, including both encoder-based and encoder-free models, and reports substantial KV-cache and FLOP reductions with relatively small quality degradation.

The empirical coverage is a clear strength, and the method is practically motivated and easy to understand. The main weakness is that the paper does not adequately isolate why LightKV works: the central claim is that prompt-aware guidance is the key differentiator, but the experiments do not directly show that this guidance signal is informative or necessary relative to simpler alternatives.

**Audience:**

Yes

**Audience Explanation:**

This is a relevant paper for researchers working on efficient multimodal inference and large vision-language models.

**Claims And Evidence:**

Yes

**Claims Explanation:**

The paper provides substantial evidence for its main empirical claim that LightKV is an effective training-free method for reducing prefill-stage vision-token KV cache in LVLMs while largely preserving performance. The evaluation spans 8 LVLMs, 8 public benchmarks, and multiple efficiency metrics, which is stronger than what is typical for this kind of systems-oriented work. The paper also evaluates both encoder-based and encoder-free models, which makes the practical scope more convincing. The reported efficiency gains are concrete rather than purely theoretical: for example, the paper reports roughly halved KV memory at 55% compression, sizable FLOP reductions, and TTFT reductions.

That said, the evidence is less convincing for the stronger mechanistic claim that prompt-aware guidance is the reason LightKV outperforms prior approaches. The key missing experiment is an ablation replacing the prompt-guidance weights with uniform weights. Without that, it is hard to tell whether the gains come from cross-modal guidance specifically, or from the hierarchical/locality-preserving compression design. Similarly, the paper does not compare against a simpler prompt-guided merging baseline, so it remains unclear whether graph message passing is essential.

I also have some reservations about the fairness of the comparisons. LightKV is tuned per model via configuration choices such as compression layers and window sizes, while at least some baselines appear to be used with more fixed settings. The discussion around FastV in particular is therefore less conclusive than stated. Overall, the paper supports the claim that the method works well empirically, but it does not yet fully support the stronger claim about why it works.

**Requested Changes:**

The following changes would strengthen the work:

- Add an ablation in which the prompt-guidance weights are replaced with uniform weights. This is the most important missing experiment because it directly tests the paper's central claim that prompt-aware guidance is responsible for the gains.

- Add a simpler prompt-guided merging baseline, or otherwise disentangle prompt guidance from graph message passing. At present, it is unclear whether the improvement comes from the weighting signal, the message-passing formulation, or the hierarchical windowing/locality prior.

- Make the FastV comparison fairer. Either tune FastV per model in a comparable way, or soften the claims that attribute the gap primarily to LightKV's architectural advantages.

---

> ### Author Response · Authors · 2026-04-04
> **Response to reviewer cQ9i**
>
> We greatly appreciate the constructive feedback the reviewer has presented, and would like to address them below.
>
> - **[Q1, Q2]: Ablation studies on the effect of prompt guidance.**
>
>     We thank the reviewers for highlighting this concern. In response, we conducted additional ablation studies in Tables 1 and 2 below to isolate the effect of prompt guidance in graph message passing with simpler baselines. Specifically, we disentangle the prompt-guidance signal by evaluating graph message passing (GMP) that utilize uniform weights and random weights. The results show a consistent and noticeable performance drop across the evaluated benchmarks when uniform or random weights are used. This verifies that the gains are not merely due to graph message passing architecture, but also stem from the prompt-guidance.
>
>     **Table 1**: Performance comparison of various prompt-guidance weighting mechanisms on LLaVA-v1.5-13B.
>
>     | **Method** | **Coco** | **MME-C** | **MME-P** | **NoCaps** | **Pope-Acc** | **Pope-F1** | **Seed** | **VizWiz** | **Avg** |
>     | --- | --- | --- | --- | --- | --- | --- | --- | --- | --- |
>     | Prompt-guided | 1.15 | 302.1 | 1543.8 | 1.08 | 0.87 | 0.86 | 0.69 | 0.56 | **99.94** |
>     | Uniform | 1.14 | 299.6 | 1535.0 | 1.06 | 0.87 | 0.86 | 0.39 | 0.55 | 93.78 |
>     | Random | 1.14 | 300.0 | 1534.5 | 1.07 | 0.87 | 0.85 | 0.39 | 0.56 | 93.96 |
>
>     **Table 2**: Performance comparison of various prompt-guidance weighting mechanisms on LLaVA-NeXT-13B.
>
>     | **Method** | **Coco** | **MME-C** | **MME-P** | **NoCaps** | **Pope-Acc** | **Pope-F1** | **Seed** | **VizWiz** | **Avg** |
>     | --- | --- | --- | --- | --- | --- | --- | --- | --- | --- |
>     | Prompt-guided | 0.96 | 326.1 | 1576.5 | 0.83 | 0.87 | 0.86 | 0.69 | 0.61 | **98.12** |
>     | Uniform | 0.98 | 311.0 | 1547.3 | 0.85 | 0.86 | 0.85 | 0.35 | 0.60 | 91.18 |
>     | Random | 0.97 | 311.1 | 1542.3 | 0.84 | 0.86 | 0.86 | 0.34 | 0.59 | 90.65 |
>
> - **[Q3]: Fairness in comparing with FastV.**
>
>     We thank the reviewer for this valuable suggestion and fully agree that fair baseline comparison is important.
>
>     In our experiments, we followed the default FastV configuration to remain identical to its original implementation. Specifically, FastV was introduced with pruning at a very early layer (specifically, layer index K=2), a key design choice of the method.
>
>     For completeness, we have conducted additional experiments by tuning FastV’s hyperparameters. Specifically, we vary the pruning layer K over [1, 4, 8], in addition to the default setting K=2 reported in the main paper, and adjust the retention ratio R accordingly so that all variants maintain the same overall KV-cache compression rate of 55%. As shown in Tables 3 and 4, LightKV consistently outperforms FastV across these different choices of K. We appreciate the reviewer for this helpful suggestion, and we will incorporate these results in our manuscript for a fairer comparison.
>
>     **Table 3**: Performance comparison of LightKV with FastV at various values of K on LLaVA-v1.5-7B.
>
>     | **Method** | **Coco** | **MME-C** | **MME-P** | **NoCaps** | **Pope-Acc** | **Pope-F1** | **Seed** | **VizWiz** | **Avg** |
>     | --- | --- | --- | --- | --- | --- | --- | --- | --- | --- |
>     | LightKV | 1.11 | 357.5 | 1519.8 | 1.03 | 0.87 | 0.86 | 0.66 | 0.53 | **99.79** |
>     | FastV K=1 | 1.08 | 337.8 | 1469.7 | 1.02 | 0.83 | 0.81 | 0.57 | 0.54 | 95.45 |
>     | FastV K=2 | 1.10 | 351.1 | 1513.7 | 1.04 | 0.85 | 0.83 | 0.66 | 0.54 | 99.03 |
>     | FastV K=4 | 1.10 | 339.2 | 1500.0 | 1.04 | 0.84 | 0.81 | 0.66 | 0.54 | 98.06 |
>     | FastV K=8 | 1.10 | 371.7 | 1502.0 | 1.02 | 0.85 | 0.84 | 0.65 | 0.53 | 99.13 |
>
>     **Table 4**: Performance comparison of LightKV with FastV at various values of K on LLaVA-NeXT-7B.
>
>     | **Method** | **Coco** | **MME-C** | **MME-P** | **NoCaps** | **Pope-Acc** | **Pope-F1** | **Seed** | **VizWiz** | **Avg** |
>     | --- | --- | --- | --- | --- | --- | --- | --- | --- | --- |
>     | LightKV | 0.98 | 338.6 | 1517.3 | 0.83 | 0.88 | 0.86 | 0.69 | 0.58 | **98.85** |
>     | FastV K=1 | 0.96 | 326.0 | 1495.2 | 0.85 | 0.86 | 0.84 | 0.68 | 0.60 | 97.87 |
>     | FastV K=2 | 0.88 | 265.4 | 1341.3 | 0.78 | 0.81 | 0.77 | 0.69 | 0.58 | 90.37 |
>     | FastV K=4 | 0.98 | 298.9 | 1504.5 | 0.86 | 0.87 | 0.85 | 0.68 | 0.59 | 97.38 |
>     | FastV K=8 | 0.96 | 293.2 | 1505.7 | 0.84 | 0.87 | 0.84 | 0.67 | 0.60 | 96.52 |

---

### Review · Reviewer_Ax8g · 2026-03-22

**Summary Of Contributions:**

The paper addresses a practical bottleneck in large vision-language models: the KV cache becomes very large during the prefill stage. Because images generate far more tokens than text, processing them requires a substantial amount of GPU memory. This is an important real-world challenge, especially in memory-constrained deployment settings. As a result, this work focuses on a practical systems challenge that matters for real deployment, not just on achieving a small benchmark improvement.

Strengths
1. A key idea of LightKV is that it does not compress image tokens using only visual similarity. Instead, it uses text prompt attention to estimate which visual tokens are more important for the current question. The model tries to keep the image regions that matter most for the prompt, rather than treating all visual tokens as equally important.

2. The method is training-free, so it can be added to existing models without retraining them.

3. The paper provides strong experimental evidence that LightKV is broadly useful, not just effective in one narrow setting. In Section 4.1, the method is evaluated on eight open-source LVLMs and eight public benchmarks, covering both open-ended generation tasks and multiple-choice reasoning tasks. In Section 4.2 and Tables 1–3, LightKV consistently keeps performance very close to the vanilla models while achieving large efficiency gains.

Although the paper presents an innovative method and strong experimental evidence, it would be helpful if the authors could also address the following weaknesses to further strengthen the paper.

Weaknesses:

1. The intra-window compression is based on a forced bipartite split (as described in Eq. 4 in Section 3.2.1, the tokens in each window are divided into two sets XA and XB), and matching is allowed only across these two sets. This is a strong approximation, because the selected pairs are not globally optimal but only optimal under the bipartite constraint. If the most similar tokens fall into the same set, they cannot be matched. Although a forced bipartite split reduces computation, it may weaken the quality of token compression.

2. Section 3.2.1 describes that the message passing and update procedure is performed independently for each window. Later, Section 3.2.2 adds inter-window compression, which allows larger-range interaction across windows. The intra-window compression may be purely local as tokens in neighboring windows cannot interact at this stage. Later inter-window compression only partly fixes the problem because the later inter-window compression is not operating on the original full set of tokens. If the intra-window step already merged/removed some tokens early, that information may already be lost. Could we have a better way to capture cross-window relations like an inter-window compression that operating on the original full set of tokens?

3. There is no direct experiment comparing bipartite matching with full matching. The paper says the bipartite design is used because it is faster and cheaper, but it does not directly test how this approximation compares with full pairwise matching inside a window. So it is unclear how much performance is lost by using this simpler bipartite, and whether the speedup is worth that trade-off.

4.  The hierarchical compression strategy (which is introduced in Section 3.2.2 Inter-window token compression) looks heuristic and manually designed. We must first decide Λ, W, and P settings for which decoder layers should do compression, how many windows to use at each stage, how much compression to apply each time. The method is not learning this schedule automatically from data and the method may work well partly because the we picked a good schedule, not because the hierarchy itself is universally strong.

5. There is no clean ablation that isolates the inter-window module alone. Without this kind of isolated experiment, the reader may ask a question: is the inter-window module really an important reason the method works better, or are most of the gains already coming from the earlier intra-window compression?

**Audience:**

Yes

**Audience Explanation:**

The paper addresses an important and practical problem in efficient LVLM inference, and its prompt-aware, training-free compression approach is likely to be of interest to researchers working on multimodal models, inference efficiency, and KV-cache optimization.

**Claims And Evidence:**

Yes

**Claims Explanation:**

The paper provides strong experimental evidence that LightKV is broadly useful, not just effective in one narrow setting. In Section 4.1, the method is evaluated on eight open-source LVLMs and eight public benchmarks, covering both open-ended generation tasks and multiple-choice reasoning tasks. In Section 4.2 and Tables 1–3, LightKV consistently keeps performance very close to the vanilla models while achieving large efficiency gains.

Although the paper provides solid experimental support, addressing the following concerns would further improve its clarity and overall strength.

1. There is no direct experiment comparing bipartite matching with full matching. The paper says the bipartite design is used because it is faster and cheaper, but it does not directly test how this approximation compares with full pairwise matching inside a window. So it is unclear how much performance is lost by using this simpler bipartite, and whether the speedup is worth that trade-off.

2. There is no clean ablation that isolates the inter-window module alone. Without this kind of isolated experiment, the reader may ask a question: is the inter-window module really an important reason the method works better, or are most of the gains already coming from the earlier intra-window compression?

**Requested Changes:**

Although the paper presents an innovative method and strong experimental evidence, it would be helpful if the authors could also address the weaknesses listed in 'Summary Of Contributions' above to further strengthen the paper.

---

> ### Author Response · Authors · 2026-04-04
> **Response to reviewer Ax8g (part 1)**
>
> We sincerely thank the reviewer for the constructive comments and suggestions, and provide our responses below.
>
> - **[W1, W3, Q1]: Choice of bipartite matching and comparison to full pairwise matching.**
>
>     We appreciate the reviewer’s insightful comment. Our choice in bipartite matching stems from two considerations: (a) method performance and (b) computational efficiency.
>
>     **(a) Method performance**: We agree that bipartite matching does not guarantee a globally optimal matching. As such, we conducted additional experiments to compare the performance between bipartite matching and full matching in Tables 1 and 2, which show marginal differences in performance. The result could be attributed to our multi-stage compression strategy. For our method, although the globally optimal pairs may fall into the same set in earlier stages (thus not matched and compressed together), they are likely to be selected into different sets in later stages, and thus matched and merged.
>
>     **Table 1**: Performance comparison of bipartite matching to full pairwise matching on LLaVA-v1.5-13B.
>
>     | **Method** | **MME-C** | **MME-P** | **Pope-Acc** | **Pope-F1** | **Avg** |
>     | --- | --- | --- | --- | --- | --- |
>     | Bipartite | 302.1 | 1543.8 | 0.87 | 0.86 | 100.75 |
>     | Full pairwise | 298.9 | 1532.0 | 0.87 | 0.86 | 100.31 |
>
>     **Table 2**: Performance comparison of bipartite matching to full pairwise matching on LLaVA-v1.5-7B.
>
>     | **Method** | **MME-C** | **MME-P** | **Pope-Acc** | **Pope-F1** | **Avg** |
>     | --- | --- | --- | --- | --- | --- |
>     | Bipartite | 357.5 | 1519.8 | 0.87 | 0.86 | 100.30 |
>     | Full pairwise | 371.0 | 1522.2 | 0.86 | 0.85 | 100.71 |
>
>     **(b) Computational efficiency**: We sincerely thank the reviewer for raising the concern on the trade-off between performance and efficiency. In Section 3.2.1 (page 6), we derive the complexity advantage of bipartite over full pairwise matching, reducing the number of comparisons from $v_w^2/2 \rightarrow v_w^2/4$. Furthermore, in Table 3 below, we report FLOPs for these two implementations at runtime. Our key observation is that given the same number of vision tokens, full pairwise matching incurs 4x FLOPs as bipartite matching. This observation can be similarly and directly extended to VRAM usage.
>
>     **Table 3**: Comparison of FLOPS between bipartite matching and full pairwise matching at 512, 1024, 2048, and 4096 vision tokens. Full pairwise matching uses 4x the number of operations as bipartite.
>
>     | **Method** | **512** | **1024** | **2048** | **4096** |
>     | --- | --- | --- | --- | --- |
>     | Bipartite | 0.134 | 0.538 | 2.151 | 8.602 |
>     | Pairwise | 0.538 | 2.151 | 8.603 | 34.410 |
>
>     Overall, combining the two factors above, we believe our method achieves a favorable trade-off between model performance and computational efficiency. We greatly appreciate the reviewer for this valuable comment.

---

> ### Author Response · Authors · 2026-04-04
> **Response to reviewer Ax8g (part 2)**
>
> - **[W2]: Design of window partitioning.**
>
>     We thank the reviewer for this insightful point on local constraints and potential information loss. We would like to address this from two aspects.
>
>     **(a)** Previous studies have verified that in LLMs and LVLMs, earlier layers capture local semantics, while later layers capture global relations [1, 2]. We therefore harness this observation and aim to design a iterative strategy (with intra- and inter-window compression) to achieve efficient vision token compression. Specifically:
>
>     - **Intra**-window compression prioritizes highly-semantic redundancy between adjacent image patches. This prevents semantically unrelated tokens from distant parts of the image being incorrectly paired. We then employ message passing to compress tokens with similar features to mitigate information loss.
>     - **Inter**-window compression expands the receptive field and focuses on capturing dependencies at longer ranges necessary. In this way, tokens from different windows in previous stages may be assigned to the same window in later stages, thus allowing features from different windows to interact at deeper layers.
>
>     **(b)** To further address this concern, we conducted additional experiments to perform compression directly on the full set of vision tokens. As observed in the second row of Tables 4 and 5, there is a noticeable degradation of overall performance. A possible reason to this is that distant tokens which are semantically unrelated are compressed sub-optimally. This highlights the need to restrict compression to a localized neighbourhood first.
>
>     **Table 4**: Performance comparison to the two compression strategies on LLaVA-v1.5-13B.
>
>     | **Strategy** | **Coco** | **MME-C** | **MME-P** | **NoCaps** | **Pope-Acc** | **Pope-F1** | **Seed** | **VizWiz** | **Avg** |
>     | --- | --- | --- | --- | --- | --- | --- | --- | --- | --- |
>     | Ours (Intra → Inter) | 1.15 | 302.1 | 1543.8 | 1.08 | 0.87 | 0.86 | 0.69 | 0.56 | **99.94** |
>     | Global-only | 1.15 | 299.6 | 1530.2 | 1.08 | 0.87 | 0.85 | 0.39 | 0.55 | 93.92 |
>     | Local-only | 1.15 | 290.0 | 1529.9 | 1.08 | 0.87 | 0.85 | 0.38 | 0.56 | 93.55 |
>
>     **Table 5**: Performance comparison to the two compression strategies on LLaVA-NeXT-13B.
>
>     | **Strategy** | **Coco** | **MME-C** | **MME-P** | **NoCaps** | **Pope-Acc** | **Pope-F1** | **Seed** | **VizWiz** | **Avg** |
>     | --- | --- | --- | --- | --- | --- | --- | --- | --- | --- |
>     | Ours (Intra → Inter) | 0.96 | 326.1 | 1576.5 | 0.83 | 0.87 | 0.86 | 0.69 | 0.61 | **98.12** |
>     | Global-only | 0.98 | 311.1 | 1549.8 | 0.85 | 0.87 | 0.86 | 0.34 | 0.60 | 91.31 |
>     | Local-only | 0.97 | 318.5 | 1543.8 | 0.85 | 0.87 | 0.86 | 0.34 | 0.60 | 91.44 |
>
>     **References:**
>
>     [1] Du, Jason, et al. "How GPT learns layer by layer." *arXiv preprint arXiv:2501.07108* (2025).
>
>     [2] Li, Bozhou, et al. "Semantic Routing: Exploring Multi-Layer LLM Feature Weighting for Diffusion Transformers." *arXiv preprint arXiv:2602.03510* (2026).
>
>
>
> - **[W5, Q2]: Ablation studies isolating intra and inter window configurations.**
>
>     We sincerely appreciate the reviewer’s comments on additional ablation studies. Following this suggestion, we conducted additional experiments to better isolate this effect by comparing to two other compression strategies: global-only, which applies directly to the original token set, and local-only. In addition to the analysis from the previous concern, the results in Tables 4 and 5 above show that both these variants underperform when compared to our strategy.

---

> ### Author Response · Authors · 2026-04-04
> **Response to reviewer Ax8g (part 3)**
>
> - **[W4]: Robustness of the schedule and the advantages of a training-free design.**
>
>     We thank the reviewer for this important comment. We agree that the schedule $\Lambda, \mathcal{W},\mathcal{P}$ is currently a heuristic design rather than one automatically learned from data. Our primary objective was to develop a **training-free**, plug-and-play solution for any LVLM without the cost of retraining or learning a policy network to decide on the compression schedule.
>
>     To ensure generalization, we first determine the schedule parameters using Coco and MME, then hold them fixed across the remaining tasks, reducing the possibility of the schedule “overfitting” to a any particular tasks. While an automatic scheduler is an interesting direction for future work, our current results demonstrate that a simple, fixed schedule still delivers favorable performance.
>
>     Additionally, we have included further sensitivity analysis in Tables 8 and 9. As presented, despite the difference schedules, the overall performance remains relatively stable.
>
>     **Table 6**: Performance comparison of varying schedules on performance on LLaVA-v1.5-13B.
>
>     | Schedule | **Coco** | **MME-C** | **MME-P** | **NoCaps** | **Pope-Acc** | **Pope-F1** | **Seed** | **VizWiz** | **Avg** |
>     | --- | --- | --- | --- | --- | --- | --- | --- | --- | --- |
>     | $\Lambda=[15,23,31], \mathcal{W}=[4,2,1]$ | 1.15 | 302.1 | 1543.8 | 1.08 | 0.87 | 0.86 | 0.69 | 0.56 | **99.94** |
>     | $\Lambda=[15,23,31], \mathcal{W}=[6,4,2]$ | 1.15 | 295.5 | 1530.8 | 1.08 | 0.88 | 0.86 | 0.69 | 0.57 | 99.92 |
>     | $\Lambda=[9,19,29], \mathcal{W}=[4,2,1]$ | 1.17 | 263.6 | 1518.9 | 1.07 | 0.87 | 0.86 | 0.69 | 0.52 | 97.32 |
>     | $\Lambda=[9,19,29], \mathcal{W}=[6,4,2]$ | 1.17 | 263.7 | 1518.9 | 1.07 | 0.87 | 0.86 | 0.68 | 0.53 | 97.38 |
>
>     **Table 7**: Performance comparison of varying schedules on performance on LLaVA-v1.5-7B.
>
>     | Schedule | **Coco** | **MME-C** | **MME-P** | **NoCaps** | **Pope-Acc** | **Pope-F1** | **Seed** | **VizWiz** | **Avg** |
>     | --- | --- | --- | --- | --- | --- | --- | --- | --- | --- |
>     | $\Lambda=[12,18,24], \mathcal{W}=[6,4,2]$ | 1.11 | 357.5 | 1519.8 | 1.03 | 0.87 | 0.86 | 0.66 | 0.53 | **99.79** |
>     | $\Lambda=[12,18,24], \mathcal{W}=[4,2,1]$ | 1.11 | 354.3 | 1512.6 | 1.04 | 0.87 | 0.85 | 0.67 | 0.53 | 99.66 |

---

> > ### Comment · Reviewer_Ax8g · 2026-04-10
> > **Thank you very much for your meticulous rebuttal**
> >
> > Thank you very much for your meticulous rebuttal and for providing solid, comprehensive experimental data to address my concerns.
> >
> > 1. The additional ablation studies and FLOPs comparisons clearly show the proposed method drastically reduces computational overhead while matching the accuracy of far more expensive baselines.
> >
> > 2. Comparing your hierarchical method against global-only and local-only baselines demonstrates that your hierarchical step-by-step strategy is practically necessary for preserving critical image information.

---

### Decision · Action_Editor_moAT · 2026-04-11

**Recommendation:** Accept with minor revision

**Additional Comments:**

The authors have conducted a detailed rebuttal that satisfactorily addressed all reviewer concerns with good experimental evidence. All three reviewers acknowledged the quality of the responses. I recommend acceptance contingent on the following minor revisions being incorporated into the camera-ready:
- Integrate the additional ablation results (bipartite vs. full matching, prompt-guidance ablation, global/local-only baselines, schedule sensitivity, similarity metrics, tuned FastV comparisons) into the main paper or appendix.
- Address Reviewer `GidE`'s notation and clarity suggestions (definition of $\mathcal{E}$, clarification of many-to-one aggregation, notation table) as outlined in the rebuttal.
- Soften claims regarding FastV comparisons per Reviewer `cQ9i`'s suggestion, incorporating the tuned FastV results for a fairer comparison.

**Audience:**

Yes

**Audience Explanation:**

KV-cache efficiency for vision-language models is a pressing practical problem, and a training-free, plug-and-play solution that works across diverse model families has clear value for both the systems and the multimodal learning communities.

**Claims And Evidence:**

Yes

**Claims Explanation:**

The submission presented solid empirical coverage across 8 LVLMs and 8 benchmarks. During the rebuttal, the authors went further and ran a substantial set of additional experiments that directly target each reviewer's concern -- bipartite vs. full matching with FLOPs breakdown, prompt-guidance ablation (uniform/random weights), hierarchical vs. global-only vs. local-only compression, schedule sensitivity, alternative similarity metrics, and a properly tuned FastV comparison. These new results are convincing and consistent with the paper's claims. All three reviewers were satisfied with the responses.